# Self-Captioning Multimodal Interaction Tuning:
# Amplifying Exploitable Redundancies for Robust Vision Language Models

**Yuriel Ryan** [1]  **Ip Hei Man** [2]  **Adriel Kuek** [2]  **Paul Pu Liang** [3]  **Roy Ka-Wei Lee** [1]

## Abstract

Current vision language models face hallucination and robustness issues against ambiguous or corrupted modalities. We hypothesize that these issues can be addressed by exploiting the shared information between modalities to compensate for the impaired one. To this end, we analyze multimodal interactions – redundant (shared), unique (exclusive), and synergistic (emergent) task-relevant information provided by the modalities – to determine their impacts on model reliability. Specifically, amplifying redundant interactions would increase this exploitable shared information to resolve these issues; yet, modern instruction datasets often eliminate redundancies to prioritize visual grounding. We bridge this gap through a self-captioning workflow featuring a MULTIMODAL INTERACTION GATE: a mechanism to convert unique interactions into redundant interactions. Our findings suggest that increasing redundancy can reduce visual induced errors by 38.3% and improve consistency by 16.8%. Code available at https://github.com/yurielryan/Multimodal-Interaction-Tuning.git

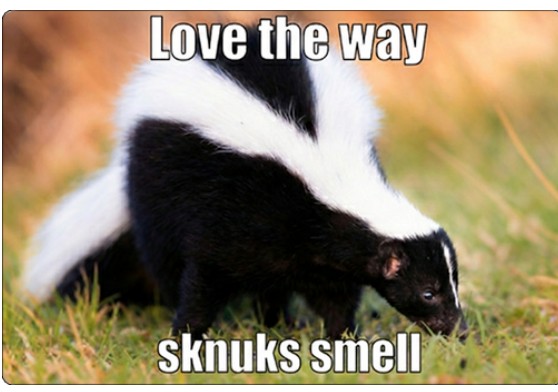

*Figure 1.* In this toy example, the modalities share a sufficient amount of (redundant) information to cover for an ambiguous text modality: the literal visual presence of the skunk provides evidence that "sknuks" is indeed a typographical error.

## 1. Introduction

Vision language models (VLMs) have achieved remarkable feats across diverse multimodal tasks. Yet, they remain prone to hallucinations and vulnerable to corrupted or ambiguous inputs. To mitigate these issues, VLMs are trained to be grounded on the visual content (Favero et al., 2024) with instruction tuning datasets (Lin et al., 2024). These datasets force the backbone language model to utilize the visual modality. While this approach has successfully trained

many families of VLMs (Yang et al., 2025a; Liu et al., 2023; Wang et al., 2025), the data mixtures in these instruction datasets remain inconsistent. This leaves data curation and augmentation strategies to intuition and heuristics, making it difficult to establish systematic approaches for more methodical progress towards robust VLMs. As such, this forms the first research question for this work:

**RQ1:** *Can a systematic data augmentation strategy, informed by insights from the data, be established to direct more intentional improvements towards robust VLMs?*

Recent works on multimodal interactions have the potential to provide the necessary insights for such a systematic approach. These interactions explain how different modalities contribute to a task (Liang et al., 2023a): redundant interactions indicate overlapping contributions; unique interactions indicate exclusive contributions from one modality; and synergistic interactions indicate a complementary structure within the dataset. Prior works have utilized these insights to train better multimodal models with experts that specialize in these interactions (Xin et al., 2025) or for better interpretability (Dewan et al., 2024). In particular, redundant interactions have been adapted into objective functions to train more robust multimodal models (Wörtwein et al., 2024; Nguyen et al., 2025). The core intuition for the robustness—defined in this work as resilience to ambiguous or degraded

---
[1]Singapore University of Technology and Design [2]DSO National Laboratories [3]Massachusetts Institute of Technology. Correspondence to: Yuriel Ryan <yurielryan@gmail.com>.

*Proceedings of the 43rd International Conference on Machine Learning*, Seoul, South Korea. PMLR 306, 2026. Copyright 2026 by the author(s).

modalities—stems from utilizing the shared (redundant) information to cover for ambiguous or corrupted modalities (See Figure 1).

However, utilizing this redundancy requires the dataset to contain exploitable redundant interactions; this is not necessarily true for all datasets. Instruction datasets in fine-tuning VLMs are designed to have less redundancy to concentrate task-relevant information in the visuals (Lin et al., 2024; Laurençon et al., 2024). In light of this visual grounding convention and the potential for redundancy to train robust VLMs, we form the second research question in this work:

**RQ2:** *Can redundant interactions be systematically increased within grounding-centric datasets to improve VLM robustness against ambiguous or corrupted modalities?*

We address the two research questions through a self-captioning workflow featuring the MULTIMODAL INTERAC-TION GATE: a mechanism to adjust interactions by transferring unique visual interactions into redundant interactions. In doing so, we simultaneously provide a framework to increase exploitable shared information in the dataset and a systematic data augmentation strategy to train more robust VLMs against ambiguous or corrupted modalities. We summarize our contributions below:

i We analyzed, through the Pointwise Partial Information Decomposition framework, how redundancies in vision language instruction datasets can produce more robust VLMs against ambiguous and corrupted modalities.

ii We operationalize our analysis through the MULTI-MODAL INTERACTION GATE: a data-centric mechanism to leverage the visual grounding convention of instruction datasets and systematically increase redundancy by transferring unique visual contributions into redundant interactions.

iii We provide empirical evidence that increasing redundant information in the training data improves a vision language model's robustness against ambiguous and corrupted inputs, reducing visual induced errors by up to 38.3% and improving consistency by 16.8%.

## 2. Related Work

**Multimodal Learning**  Vision Language Models are often improved by visually grounding them with data augmentation strategies (Liu et al., 2023; Deng et al., 2024). Recent works expanded on this paradigm to address hallucination and robustness issues (Favero et al., 2024; Li et al., 2025b; Zou et al., 2025; Zhao et al., 2025); yet, these issues remain persistent (Chen et al., 2023; Geigle et al., 2024; Guan et al., 2024). This motivates us to re-evaluate the heavy emphasis on visual grounding in instruction datasets.

**Multimodal Interactions**  Partial Information Decomposition (Williams & Beer, 2010) offers a framework for analyzing the interactions between sources of information (Bertschinger et al., 2014; Ince, 2017; Finn & Lizier, 2018; Wollstadt et al., 2023). This framework has been adapted into multimodal machine learning to quantify interactions between modalities (Liang et al., 2023a; 2024; Yang et al., 2025b), providing insights to improve multimodal models. For instance, redundant interactions—shared information between modalities—were incorporated into objective functions to improve robustness (Wörtwein et al., 2024; Nguyen et al., 2025) while unique interactions—modality exclusive task-relevant information—indicate dominant modalities (Liang et al., 2023b). These insights have also been utilized for various applications such as interpretability (Dissanayake et al., 2025; Wenderoth et al., 2025; Dewan et al., 2024; Xin et al., 2025).

Unlike prior works, we adjust the interactions in the dataset instead of merely utilizing them; specifically, we amplify redundant interactions to increase shared information for more robust VLMs against ambiguous or corrupted modalities.

## 3. Multimodal Interactions

We analyzed multimodal interactions for robust VLMs. Specifically, we examine redundant interactions to improve robustness against ambiguous or corrupted modalities.

### 3.1. Preliminaries

**Partial Information Decomposition (PID)**  Proposed by Williams and Beer (2010), PID describes how multiple sources, $X_1$ and $X_2$, provide information about a third variable $Y$. This framework decomposes information from the inputs into shared, unique, and synergistic contributions:

$$I(X_1, X_2; Y) = R + U_1 + U_2 + S$$
$$I(X_1; Y) = R + U_1 \quad I(X_2; Y) = R + U_2$$

$I(X_1, X_2; Y)$ denotes the total amount of information that $X_1$ and $X_2$ provide about $Y$. $R$ represents the redundant (shared) information from the inputs while $U_1$ and $U_2$ represents the unique contribution by $X_1$ and $X_2$ respectively. Synergy, represented by $S$, is the emergent information in jointly observing $X_1$ and $X_2$. We provide visualizations[1] of these interactions in Figure 2.

In multimodal data, estimators—PID-Batch (Liang et al., 2023a) and LSMI (Yang et al., 2025b)—represent individual modalities as $X_1$ and $X_2$ to quantify how individual modalities could interact with each other to produce task-relevant information. In the following paragraphs, we further define the notations $R$, $U$, and $S$ in the vision-language setting.

---

[1]The image in Figure 2 is for research purposes and is sourced from here; all credits remain with the original authors.

**Redundant** Interactions

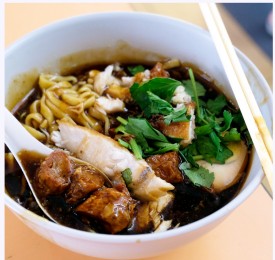

The image is a picture of a popular noodle dish: Lor Mee

**Captions**

**Unique** Interactions

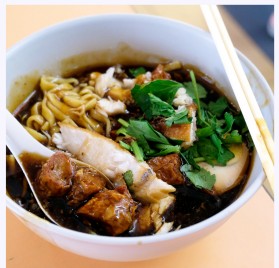

Does the image contain food? Is the item in the image edible? Is this safe for consumption?

**Prompt Ensembles**

**Synergistic** Interactions

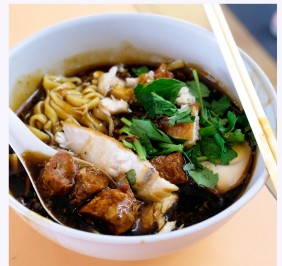

What is the name of this dish and is it commonly found in Singapore?

**Reasoning**

*Figure 2.* Illustrations of $R$, $U$, and $S$ in multimodal data. Redundant interactions are commonly observed in captioning tasks where the text shares the same information with the image (e.g., the noodle dish "Lor Mee" and its corresponding image). Unique interactions are commonly observed in prompt ensembling techniques or modality grounding datasets; in this example, the task-relevant information is concentrated in the image — and therefore have high $U_V$ — while the text contain paraphrased versions of the original prompt. Synergistic interactions are commonly observed in reasoning related tasks which require both the image and the text to obtain the correct answer.

**Redundant Interactions** $R$ between the text and visual modalities refer to the shared information provided by $X_T$ and $X_V$ about $Y$. This contribution is commutative (Finn & Lizier, 2018), and is represented as:

$$R = I_{red}(X_V, X_T; Y) = I_{red}(X_T, X_V; Y)$$

**Unique Interactions** $U$ represent the exclusive information an individual modality provides about $Y$:

$$U_V = I(X_V \backslash X_T; Y)$$
$$U_T = I(X_T \backslash X_V; Y)$$

$U_m$ is the unique information from modality $m$.

**Synergistic Interactions** $S$ can occur when both modalities are observed together. Unlike $R$, synergy requires both modalities to be present, and the resultant information is emergent; it cannot be found in either modality alone:

$$S = I_{syn}(X_T, X_V; Y)$$

**Sample-level Interactions** are expressed in their corresponding lower-case characters:

$$u_V = i(x_V \backslash x_T; y), \quad u_T = i(x_T \backslash x_V; y)$$
$$r = i_{red}(x_V, x_T; y), \quad s = i_{syn}(x_V, x_T; y)$$
$$i(x_V, x_T; y) = r + u_V + u_T + s$$
$$i(x_V; y) = r + u_V, \quad i(x_T; y) = r + u_T$$

**3.2. Impacts of Redundant Interactions**

In this section, we build on the theoretical foundations of Point-wise Partial Information Decomposition (PPID) (Ince, 2017; Finn & Lizier, 2018).

### 3.2.1. POINT-WISE INFORMATION

The point-wise information $i(x_m; y)$, as defined by PPID, expresses the information a modality $x_m$ provides about the task $y$ on the sample-level. This term is first decomposed (details in Appendix A) into two components:

$$i(x_m; y) = h(x_m) - h(x_m|y), \; m \in \{V, T\}$$
$$i(x_m; y) = i^+(x_m; y) - i^-(x_m; y)$$

With this decomposition, we define the following terms to quantify how informative $x_m$ is during training:

**Point-wise Specificity**: $i^+(x_m; y) = h(x_m)$. This term describes the potential for the modality $x_m$ to be informative— the rarer $x_m$ is in the dataset, the more specific it is.

**Point-wise Ambiguity** $i^-(x_m; y) = h(x_m|y)$. This term describes the potential for the modality $x_m$ to be misinformative—the remaining surprise $x_m$ provides after knowing $y$.

Building on these two non-negative terms ($i^+$ and $i^-$), we define the corresponding specificity and ambiguity terms for point-wise redundant interactions.

**Definition 3.1. Redundant Specificity** $r^+$ describes the lower bound potential for the multimodal sample to *inform* about the task; it is defined by the minimum point-wise specificity of the two modalities:

$$r^+(x_V, x_T; y) = \min_{m \in \{V, T\}} i^+(x_m; y)$$

**Definition 3.2. Redundant Ambiguity** $r^-$ describes the lower bound potential for the multimodal sample to *misin-*

*form* about the task; it is defined by the minimum point-wise ambiguity of the modalities:

$$r^-(x_V, x_T; y) = \min_{m \in \{V,T\}} i^-(x_m; y)$$

**Definition 3.3. Point-wise Redundancy** is further defined by combining $r^+$ and $r^-$ to describe the shared task-relevant information between the two modalities:

$$r = r^+(x_V, x_T; y) - r^-(x_V, x_T; y)$$

### 3.2.2. EFFECTS OF REDUNDANCY IN TRAINING VLMS

From the definition of point-wise redundancy, increasing redundant interactions in instruction datasets must either increase $r^+$, decrease $r^-$, or both. In light of this, we outline three hypotheses below that motivate our experiments in Sections 5.2 and 5.3.

**Hypothesis 1.** *Increasing $R$ in the training data induces the model to use both modalities more regularly.*

We posit that increasing redundant interactions provide opportunities to increase $r^+$. This increases the potential of the weaker modality to inform about $y$ (**Definition** 3.1), providing opportunities for the model to learn to utilize information from both modalities.

**Hypothesis 2.** *Increasing $R$ in the training data induces the model to be more consistent.*

We posit that increasing redundant interactions ($r$) provide opportunities to reduce $r^-$. This reduces the potential of the modalities to misinform the model (**Definition** 3.2), providing clearer signals during training to produce more consistent responses.

**Hypothesis 3.** *Increasing $R$ reinforces a VLM's resilience against ambiguous or corrupted inputs.*

Building on hypotheses 1 and 2, we posit that the net effects of increasing $R$—using both modalities more regularly to produce consistent outputs—produces a more resilient system against ambiguous or corrupted inputs.

## 4. Method

In the previous section, we analyzed the role of redundant information in VLM training. This motivates the MULTI-MODAL INTERACTION (MI) GATE to increase redundant interactions. While this method is limited to discrete tasks, we extensively evaluate its core ideas—interaction transfer and its mechanism—in Section 5 in the general augmentation setting.

### 4.1. Multimodal Interaction (MI) Gate

We present the MI GATE in three stages: first, we describe how interactions can be transferred; second, we detail how the interactions can be estimated for this transfer to occur; third, we build on the first two stages for the mechanism behind the MI GATE.

#### 4.1.1. INTERACTION TRANSFER

We posit that redundant information can be increased by transferring unique contributions from one modality into the shared space. This is achieved by holding the mutual information of the primary modality $I(X_1; Y)$ constant and modifying the second modality $X_2$ to encompass information from $X_1$; thus, increasing the amount of overlapping information which would increase redundancy.

In practice, we implement the transfer by captioning the image, capturing information from the image (without modifying the visuals) and adding it to the text modality. This converts exclusive visual information—previously found only in the image—into information shared by both modalities (Figure 3, Interaction Transfer).

There are two potential errors in this transfer: the captioning model could produce erroneous captions; the added captions might unintentionally increase unique text information instead of increasing redundant information. To mitigate these errors, we implement a mechanism, detailed in the following section, to control the captioning procedure. We investigate the extent of these potential errors in Section 5.1.

#### 4.1.2. ESTIMATING MULTIMODAL INTERACTIONS

To implement and validate interaction transfers, we have to first quantify these interactions. To this end, we build on an estimator based on PPID (Yang et al., 2025b) using classifiers[2] ($P_\Theta$) and entropy estimators[3] ($H_\Theta$). We utilize $H_\Theta$ and $P_\Theta$ to estimate $i^\pm(x_{m \in \{V,T\}}; y)$ to approximate point-wise redundant specificity $r^+$ and ambiguity $r^-$ (**Definitions** 3.1 and 3.2). With $r^+$ and $r^-$, we compute point-wise redundant interactions $r$ (**Definition** 3.3). Subsequently, $u_V$, $u_T$, and $s$ are derived by subtracting from their respective point-wise information terms. Note that in Algorithm 1, $r, u_V, u_T, s$ are vectors of size $N \times 1$ representing the point-wise interactions for each sample. To recover the $R, U_V, U_T$ and $S$, we take the expectation of these point-wise values. Further details in Appendix B.

#### 4.1.3. MECHANISM

The MI GATE operates by selecting a set of samples where $u_V$ is the dominant interaction ($S_{valid}$). These samples are captioned to convert $u_V$ into $r$. We control this transfer through a threshold ($\tau$) for the percentage of samples cap-

---

[2]The uni-modal and multimodal classifiers are separately trained with a 3-layer Multilayer Perceptron.

[3]We trained the entropy estimators with KNIFE (Pichler et al., 2022): a differentiable estimator using gaussian mixture models.

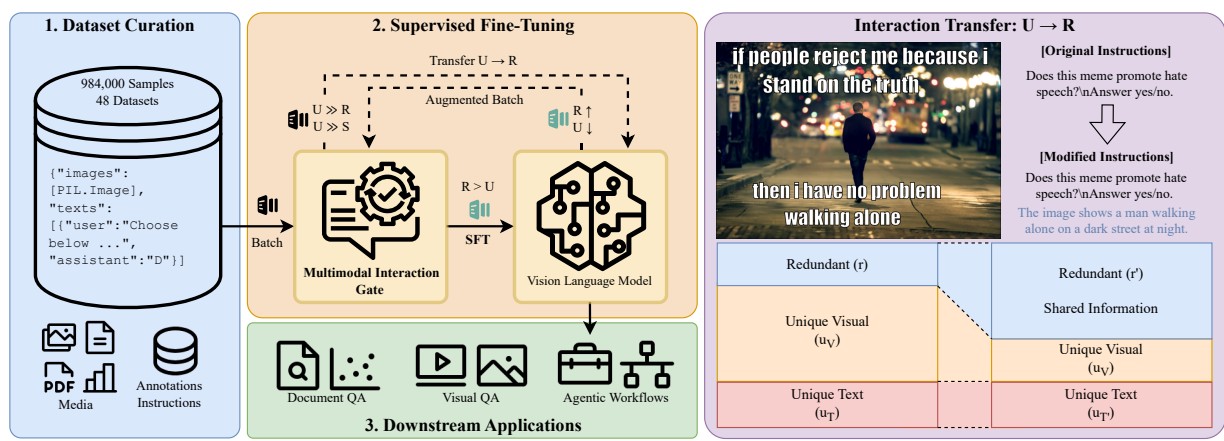

*Figure 3.* The Self-Captioning Multimodal Interaction Tuning workflow to increase exploitable redundant interactions. This workflow utilizes the MULTIMODAL INTERACTION (MI) GATE to systematically filter samples with high unique visual information to be captioned by the VLM, transferring the unique visual information into shared (redundant) information prior to the training loop.

---

**Algorithm 1** Estimate Interactions $(\mathcal{D}, \mathcal{F})$

**Require:** Dataset $\mathcal{D} = \{(x_V, x_T, y)_n\}_{n=1}^N$;
**Require:** Embedding Model $\mathcal{F}(\mathcal{D})$
1: $X_V, X_T \leftarrow \mathcal{F}(\mathcal{D})$
2: Train entropy estimators: $H_\theta$
3: Train unimodal classifiers: $P_\theta(Y|X_V), P_\theta(Y|X_T)$
4: Train multimodal classifier: $P_\theta(Y|X_V, X_T)$
5: Let $X_m$ be $X_V, X_T$, or both $(X_V, X_T)$
6: $i^+(X_m; Y) \leftarrow H_\theta(X_m)$
7: $i^-(X_m; Y) \leftarrow H_\theta(X_m) + \log P(Y) - \log P_\theta(Y|X_m)$
8: $r^\pm(X_V, X_T; Y) = \min_m i^\pm(X_m; Y)$
9: $\mathbf{r} = r^+(X_V, X_T; Y) - r^-(X_V, X_T; Y)$
10: $\mathbf{u}_V = i(X_V; Y) - r$
11: $\mathbf{u}_T = i(X_T; Y) - r$
12: $\mathbf{s} = i(X_V, X_T; Y) - r - u_V - u_T$
13: **Output:** $r, u_V, u_T, s$

---

**Algorithm 2** MI Gate$(\mathcal{D}, \mathcal{F}, \tau)$

**Require:** Dataset $\mathcal{D} = \{(x_V, x_T, y)_n\}_{n=1}^N$;
**Require:** Embedding Model $\mathcal{F}(\mathcal{D})$
**Require:** Threshold (%) of samples to caption $\tau$
1: $r, u_V, u_T, s \leftarrow$ Estimate Interactions $(\mathcal{D}, \mathcal{F})$
2: $\mathcal{S}_{valid} \leftarrow \{n \mid u_{V,n} = \max(r_n, u_{V,n}, u_{T,n}, s_n)\}$
3: $k \leftarrow \min(\lfloor \tau N \rfloor, |\mathcal{S}_{valid}|)$
4: Let $\mathcal{S}_{caption} \subseteq \mathcal{S}_{valid}$ such that $|\mathcal{S}_{caption}| = k$
5: $c_n \leftarrow \text{Caption}(x_{V,n}) \quad \forall n \in \mathcal{S}_{caption}$
6: $x'_{T,n} \leftarrow \text{Concat}(x_{T,n}, c_n) \quad \forall n \in \mathcal{S}_{caption}$
7: $x'_{T,n} \leftarrow x_{T,n} \quad \forall n \notin \mathcal{S}_{caption}$
8: $\mathcal{D}' \leftarrow \{(x_V, x'_T, y)_n\}_{n=1}^N$
9: **Output:** Augmented Dataset $\mathcal{D}'$

---

tioned in the dataset; this facilitates a systematic decrease in unique visual information to increase redundant interactions in the dataset as outlined in Algorithm 2.

The MI GATE can be configured to mitigate the two potential errors—erroneous captions and unintentional increases in $u_T$—in interaction transfers. We propose two hypotheses to address each error and evaluate them in Section 5.1.

**Hypothesis 4.** *Increasing the percentage of samples captioned averages out noise from erroneous captions.*

We posit that the captions, on average, would contain meaningful and largely redundant information from the image.

**Hypothesis 5.** *Captioning samples with high synergy increases $U_T$; these samples should not be captioned.*

We posit that adding captions to samples with high synergy

would increase $u_T$ instead of $r$. This is motivated by the definition of synergy as the emergent information when both modalities are jointly observed; the added captions would disrupt this structure and increase the impact of the text modality, leading to a rise in $u_T$.

### 4.2. Self-Captioning Supervised Fine-Tuning

We incorporate the MI GATE into the supervised fine-tuning workflow to systematically increase redundant interactions in the training data. This system employs the VLM to caption valid samples prior to the training loop (Figure 3).

To set redundant interactions as the independent variable and ensure reproducibility, we trained VLMs in the same family with open weights and open data. This eliminates confounding factors due to differences in parametric knowledge and attribute the results to changes in redundancy. To this end, SmolVLM (Marafioti et al., 2025) and LLaVA-OneVision-1.5 (Li et al., 2025a) were trained for task-specific and general settings in our experiments.

**Supervised Fine-tuning.** For specific tasks, we finetuned the instruct variants of the VLMs with the self-captioning workflow (Figure 3) and systematically increase redundant information with the MI GATE in the training data.

For general tasks, we finetuned the base VLMs on instruction datasets. While we are unable to fully utilize MI GATE on open-ended tasks, we increase the amount of redundant information by captioning 25% and 50% of the entire dataset. We ensured that this proportion is consistent across all categories and datasets in our training data.

The supervised fine-tuning adopts standard practices by attaching Low Rank Adapters (Hu et al., 2022) to the backbone models. We mainly used the Cauldron (Laurençon et al., 2024) instruction dataset and stratified the data with a temperature-based sampling strategy to represent each category and dataset (see Appendix C for further details). We prepared three versions of Cauldron for our experiments: a baseline set without modifications; a set with 25% of the samples with captions; a set with 50% of the samples with captions, including those from the 25% variant.

## 5. Experiments

In this section, we evaluate the core ideas for the MI GATE in the general setting and provide support for the hypotheses in Sections 3 and 4.

### 5.1. Interaction Transfer

We compared the estimated multimodal interactions before and after the inclusion of the captions across the entire dataset of Hateful Memes (Kiela et al., 2020). Below, we discuss our findings:

**Is interaction transfer valid?** From Table 1, there is substantial evidence to support the transfer of unique ($U_V$) to redundant ($R$) interactions. The training split exhibits the most substantial changes with $R$ increasing by 319.3% and $U_V$ decreasing by 50.6%. This suggests that the model is exposed to more redundant information—as a result of the modifications made—during training,

Further, the multimodal interaction estimator generalizes effectively to unseen data; the validation and test splits show a significant (109%) increase in $R$ alongside a notable (27%) drop in $U_V$, aligning with the shifts in the training set.

Furthermore, the added captions contain largely redundant information. A non-zero value for $u_V$ would signal the addition of noise from erroneous captions; yet, $U_V$ converged to 0 while $R$ increases across all splits and model sizes, suggesting that the captions—on average—are redundant. Thus, captioning on a larger scale mitigates potential erroneous captions, providing support for **Hypothesis** 4.

*Table 1.* Comparison of redundant and unique interactions on the Hateful Memes dataset (Kiela et al., 2020) before (baseline) and after modifications to the text modality across different VLMs. Values are absolute scores; percentage change in parentheses relative to the Baseline (data before modifications).

| Model | $R$ | $U_V$ | $U_T$ | $S$ |
|---|---|---|---|---|
| | | Training Set | | |
| Baseline | 0.0553 | 0.3465 | $-0.0125$ | 0.0000 |
| Random Text | 0.0682 (+23%) | 0.3380 (-2%) | 0.0000 | 0.0513 |
| SmolVLM-2B | 0.1897 (+243%) | 0.1964 (-43%) | 0.0000 | 0.0834 |
| Qwen2.5 32B | 0.2319 (**+319%**) | 0.1710 (**-51%**) | 0.0000 | 0.0857 |
| | | Validation Set | | |
| Baseline | 0.0443 | 0.2128 | $-0.0010$ | 0.0000 |
| Random Text | 0.0628 (+42%) | 0.1964 (-8%) | $-0.0249$ | 0.0000 |
| SmolVLM-2B | 0.0620 (+40%) | 0.1956 (-8%) | 0.0000 | $-0.0278$ |
| Qwen2.5 32B | 0.0925 (**+109%**) | 0.1642 (**-23%**) | 0.0000 | $-0.0302$ |
| | | Test Set | | |
| Baseline | 0.0500 | 0.1734 | $-0.0078$ | 0.0000 |
| Random Text | 0.0653 (+31%) | 0.1574 (-9%) | $-0.0269$ | 0.0000 |
| SmolVLM-2B | 0.0752 (+50%) | 0.1557 (-10%) | 0.0000 | $-0.0031$ |
| Qwen2.5 32B | 0.0939 (**+88%**) | 0.1268 (**-27%**) | 0.0000 | $-0.0332$ |

**What drives interaction transfers?** Semantic content is the primary driver of interaction transfer. Comparisons between captions and random text (Table 1) show that semantic value amplifies the transfer. Notably, $R$ rises by up to 88% and $U_V$ decreases by up to 27% when the added text is meaningful. Conversely, the negative values in $U_T$ for random text indicate that meaningless additions confused the model since it is forced to associate (meaningless) garbled letters with the image.

Notably, the increase in $R$ with SmolVLM-2B as the captioning model is more significant than random text. This provides substantial support that increasing the percentage of samples captioned—even with a much smaller model that would likely make more errors—could average out noise from erroneous captions (**Hypothesis** 4).

**Is bi-directional transfer feasible?** Bi-direction interaction transfer is feasible. In a separate experiment, we generated images, conditioned on the text modality of DocMSU (Du et al., 2024), with GPT Image 1.5. From Figure 4, we show that image generation models could successfully capture the unique information from the text, effectively converting $U_T$ to $R$. There are three important implications for this result: first, the reverse direction of transferring interactions, $U_T$ to $R$, is feasible through diffusion models; second, this transfer reinforces the claim that interaction transfers are driven by semantic content; third, interaction transfer applies to other modalities — such as audio which is frequently generated with diffusion models. We document further details in Appendix B.2.

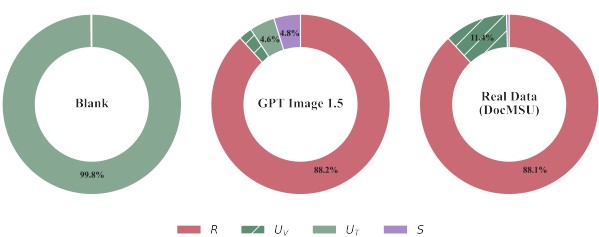

*Figure 4.* Comparison of interactions between synthetically produced (blank) images, (diffusion-based) generated images, and the real data form DocMSU (Du et al., 2024). The generated images successfully converts $U_T$ to $R$ and could even (closely) match the interaction distribution of the real data.

**Does the size of the captioning model matter?** As reflected in Table 1, the size of the captioning model affects the degree of change in the interactions. We reason that this is primarily due to the model's abilities in extracting content from the image. A smaller model is naturally expected to perform worse than a larger model. That said, the trends—increase in $R$ and decrease in $U_V$—remain consistent despite the differences in size.

Further experiments in Section 5.3 suggest that a smaller captioning model (SmolVLM-2B) was still able to improve a VLM's robustness against corrupted modalities. Figure 6 supports this trend: performance stability rises as the percentage of data captioned by SmolVLM-2B increases.

**When would interaction transfer fail?** We experimented with a forceful transfer of interactions with the synergy (S) dominant **UR-FUNNY** (Hasan et al., 2019) dataset. We captioned the videos with Qwen2.5-VL-32B-Instruct in the dataset and added it to the text modality. From Table 2, we observe a 750% increase in unique text information ($U_T$); instead of increasing redundancy, the captions replaced synergy with $U_T$ as the dominant interaction (Table 2). As a result, these observations support **Hypothesis** 5.

For datasets with high synergy, preserving this structure is vital for the VLM to learn the interdependence between the modalities. Thus, we configured the MULTIMODAL INTERACTION GATE to bypass the captioning module for these specific samples. In doing so, the model is provided opportunities to learn the synergistic structures in the dataset while simultaneously exploiting redundant information from the remaining augmented batches.

### 5.2. Robustness Against Ambiguity

We investigate the robustness of VLMs against ambiguous inputs. We build on HallusionBench (Guan et al., 2024) to characterize the mistakes into mixed errors, language induced (LI) and visual induced (VI) hallucinations. Mixed errors are mistakes made that are attributed to both modal-

*Table 2.* Change in multimodal interactions before and after captioning the videos in the **UR-FUNNY** dataset. The results reported are from the unseen test set.

| Component | Before | After | $\Delta$ |
|---|---|---|---|
| $R$ Redundancy | 0.0000 | 0.0000 | 0.0% |
| $U_V$ Unique (Visual) | 0.0003 | 0.0003 | 0.0% |
| $U_T$ Unique (Text) | 0.0004 | **0.0034** | **+750.0%** |
| $S$ Synergy | 0.0043 | **0.0030** | **-30.2%** |

*Table 3.* Macro average change in accuracy, error types, and consistency scores across relative to the baseline ($0\%\Delta R$). Positive values for LI, VI and Mixed errors indicate that the number of errors have increased; negative values indicate a decrease in errors.

| $\Delta R$ | $\Delta Acc\uparrow$ | $\Delta LI\downarrow$ | $\Delta VI\downarrow$ | $\Delta Mix\downarrow$ | $Consist.\uparrow$ |
|---|---|---|---|---|---|
| | | SmolVLM: 256M, 500M, 2B | | | |
| 25% | +2.7% | +9.5% | -23.6% | +30.3% | +8.5% |
| 50% | **+4.0%** | +15.2% | **-38.3%** | +42.9% | **+16.8%** |
| | | LLaVA-OneVision: 4B, 8B | | | |
| 25% | +2.4% | +2.9% | **-34.4%** | +23.1% | **+6.2%** |
| 50% | **+2.5%** | -6.8% | -6.5% | +4.8% | +5.5% |

ities. LI and VI hallucinations are mistakes made due to ambiguous text and visual inputs respectively. The mistakes are characterized by comparing model responses against the control groups (See Appendix D for details).

**Does increasing $R$ improve robustness against ambiguity?** Training VLM with increased redundant interactions reinforces its resilience against ambiguous inputs and made its responses more consistent. This can be seen in Table 3 with a increases in accuracy (by up to 4%) and consistency scores (by up to 16.8%) in the model families and parameter sizes. This means that the VLMs are not only making fewer mistakes against misleading or ambiguous inputs, but they are also more reliable in their outputs. These results provides support for **Hypothesis** 2: increased redundant interactions teaches a model to be more consistent as observed with the higher consistency and accuracy scores.

**How does increased $R$ influence the type of mistake?** Increasing $R$ reduces visual induced (VI) hallucinations for all model sizes. From Table 3, VI decreased by up 38.3% for SmolVLM and 34.4% for LLaVa models. Conversely, language induced (LI) and mixed errors gradually increased. These shifts have two important implications: first, there is a tradeoff—at the cost of higher LI and mixed errors—in reducing VI hallucinations; second, the VLM uses both modalities more regularly as observed with the rise in LI and mixed errors (mistakes attributed to language or both modalities), providing support for **Hypothesis** 1.

Overall, models trained with increased $R$ are more resilient

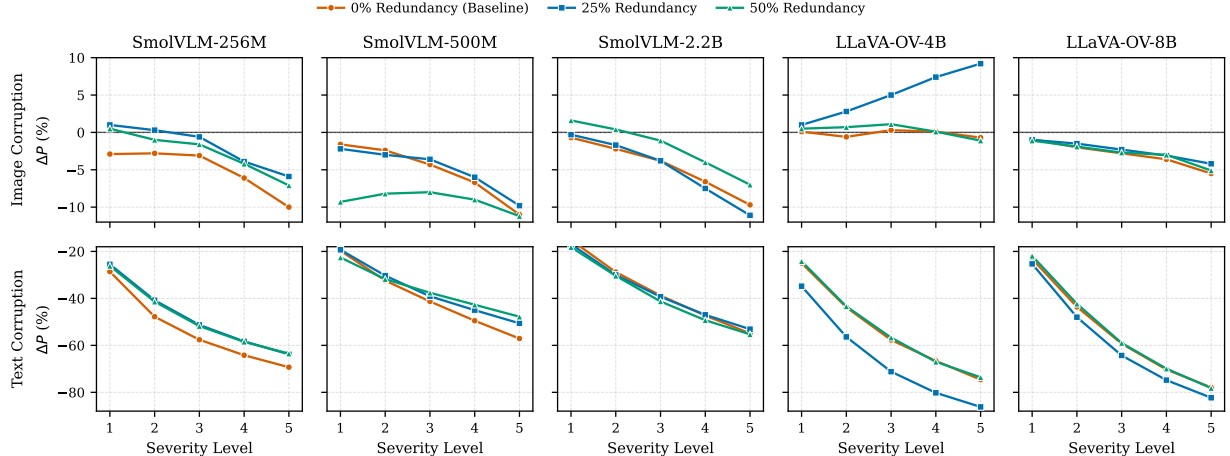

*Figure 5.* The performance stability, $\Delta P$, of SmolVLM and LLaVa-OneVision plotted against increasing levels of corruption severity for both the visual and text modalities. Generally, models trained with increased redundancies (either 25% or 50%) have more stable performance compared to the baseline models (0% additional redundancy). Absolute values in Appendix D.2, Table 11.

against ambiguous inputs. This can be seen from the net improvements in accuracy—making less mistakes due to ambiguous inputs—over the baseline model (Table 3); thus, partially supporting **Hypothesis** 3: increasing $R$ increases a VLM's resilience to ambiguous inputs.

**How do these results align with our method of transferring interactions?** We demonstrated that the captions transferred $U_V$ to $R$ in Section 5.1. In this regard, the trends in Table 3 align closely with this transfer of interactions. As $R$ increase, the model learned to use both modalities more regularly (**Hypothesis** 1); resulting in more mixed and language induced errors. As $U_V$ decreases, the model depend less on the image; decreasing the number of visual induced hallucinations.

### 5.3. Robustness Against Modality Corruption

We investigate the robustness of VLMs against corrupted modalities. We compute the performance stability, $\Delta P$, when the input data—from the GQA (Hudson & Manning, 2019) dataset—undergoes controlled levels of degradation. This corruption includes 5 levels of severity: 1 being the least severe and 5 being the most severe.

**Performance Stability** is evaluated with the relative change in performance:

$$\Delta P = \frac{P_{Corrupted} - P_{Clean}}{P_{Clean}}$$

$\Delta P \geq 0$ indicates a robust model while $\Delta P < 0$ indicates performance degradation with corrupted inputs.

Following previous works (Chen et al., 2023), we progressively corrupt the images with the addition of gaussian, impulse, and shot noise. The text modality is corrupted through

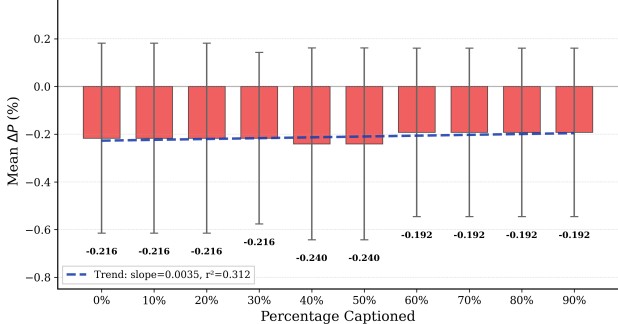

*Figure 6.* An increasing trend of $\Delta P$ as the percentage of samples captioned ($\tau$) increases in hate speech detection (Kiela et al., 2020).

random character insertions, drops, or replacements.

**Does increasing $R$ improve robustness against modality corruption?** From Figure 5, training with increased $R$ tends to have higher $\Delta P$ over the baseline model (absolute values in Appendix D, Table 11). This is consistent in corrupting either modality across all model sizes, suggesting that increasing $R$ produces more robust VLMs against degraded modalities; thus, partially supporting **Hypothesis** 3.

Notably, mild image corruption occasionally outperforms clean performance. This supports **Hypothesis** 2: increasing $R$ suppresses $r^-$, guiding the model to utilize clearer text signals when the image is ambiguous. However, a tradeoff can be observed in Figure 5 as these models struggle under text corruptions. These results largely follow the findings in Section 5.2: increasing $R$ reduces visual errors but increases language induced errors as a result of using both modalities more regularly (**Hypothesis** 1).

*Table 4.* General performance across SmolVLM and LLaVa-OneVision trained at varying redundancy levels (percentage of samples captioned).

| Model | +R | MMMU↑ | MMStar↑ | MathVista↑ | TextVQA↑ |
|-------|-----|-------|---------|-----------|----------|
|       | 0%  | 21.2  | 27.9    | 17.0      | 38.4     |
| 256M  | 25% | 22.4  | 21.9    | 21.3      | 21.9     |
|       | 50% | 20.2  | 23.3    | 18.9      | 25.3     |
|       | 0%  | 27.3  | 27.5    | 24.4      | 34.6     |
| 500M  | 25% | 25.6  | 27.0    | 22.9      | 31.8     |
|       | 50% | 25.9  | 26.7    | 23.7      | 28.7     |
|       | 0%  | 29.8  | 34.0    | 25.9      | 50.3     |
| 2B    | 25% | 32.7  | 35.6    | 25.5      | 46.8     |
|       | 50% | 34.0  | 35.6    | 26.3      | 51.2     |
|       | 0%  | 30.0  | 37.0    | 14.7      | 51.5     |
| 4B    | 25% | 40.7  | 39.9    | 8.0       | 53.8     |
|       | 50% | 34.0  | 37.7    | 14.5      | 54.2     |
|       | 0%  | 41.4  | 44.1    | 16.7      | 43.0     |
| 8B    | 25% | 49.9  | 47.0    | 32.7      | 27.2     |
|       | 50% | 42.0  | 47.9    | 35.0      | 42.6     |

**Does scaling $R$ improve performance stability?** Figure 6 suggests that scaling redundant information increases a VLM's resilience to modality corruption. In this experiment, we implemented the self-captioning workflow (Figure 3) to finetune 11 variants of SmolVLM-2B-Instruct (setting $\tau$ at 0-100% at 10% intervals) with the same model being used to caption the images. We observe a general increasing trend of $\Delta P$ which provides partial support for **Hypothesis** 3.

However, the model trained on $\tau = 100\%$ suffered catastrophic performance degradation ($\Delta P \approx -4.07$). This has an interesting implication: the training set must contain samples with a variety of interactions for the model to generalize effectively; this conclusion aligns with our study on when interactions would fail (Section 5.1). Only including redundant interactions in the training data provides no opportunities for the VLM to learn other interactions.

**Does $R$ affect general performance?** We evaluated the general VLMs on four datasets (Table 4): MMMU (Yue et al., 2024), MMStar (Chen et al., 2024), Mathvista (Lu et al., 2024), TextVQA (Singh et al., 2019). From Table 4, the general benchmarks suggests that additional redundant interactions do not have a clear impact on visual grounding. While we observe some deterioration in various benchmarks (8B LLaVa on TextVQA), training on redundant information could improve on others (8B LLaVa on MathVista). We posit that the type of task—and the percentage of samples captioned within those tasks—could affect general performance in those categories due to a variety of interactions. Future works could investigate in detail the effects of redundancy on different categories of tasks with ablations on model sizes and architectures.

## 6. Discussion and Conclusion

In this work, we address the heuristic-based approaches in curating instruction datasets (**RQ1**) by proposing an alternative: a self-captioning framework to adjust multimodal interactions with a MULTIMODAL INTERACTION (MI) GATE. This mechanism systematically increases shared information between the modalities by converting unique visual interactions into redundant ones (**RQ2**). Through our experiments, we validate the core ideas of MI GATE in the general setting with **Hypotheses** 1–5. We demonstrate that increased redundancy improves a VLM's robustness against ambiguous or corrupted modalities: visual induced hallucinations decreased by 38.3%; consistency increased by 16.8%; and a largely improved performance stability under degraded modalities. While there are net benefits (better performance against ambiguous modalities), our findings also suggest a tradeoff in a slight increase in text-based errors. Overall, this work offers a potential solution to the two research questions. Future works could further consider the impacts of multimodal interactions, moving beyond heuristics towards methodical (interaction-aware) designs for more intentional VLM improvements.

**Limitations & Future Work.** Our method transfers interactions in one direction: from the visual to the textual modality. We accept this limitation as vision-language datasets are inherently vision-centric; attempting the reverse transfer—from text to vision—would be unreliable as the text primarily serves as instructions. We leave the possibility of bi-directional transfers to future work.

Our estimator relies on neural networks limited to discrete tasks; the derived interaction values are approximations and are extrapolated to general tasks. To address this, we focus on empirical trends in our experiments and control potential confounding factors—such as training parameters—to attribute observed shifts to changes in interactions.

Finally, this work only focuses on visual and textual modalities. Despite this limitation, we offer two fundamental contributions in this work: we show that an interaction transfer is feasible; we provide evidence that redundant interactions can produce robust multimodal models. Future work could build on these findings to include more modalities.

## Acknowledgements

This research is supported by the National Research Foundation, Singapore under its AI Singapore Programme (AISG Award No: AISG3-AMP-2024-08-001). YR expresses his appreciation to his cat, Lor Mee, for serving as a patient audience and for inspiring the concept behind Figure 2.

## Impact Statement

This paper presents work whose goal is to improve the robustness of Vision Language Models against corrupted and ambiguous modalities by adjusting the multimodal interactions in the training data. While the experiments in this work uses a dataset with potential harmful content (Hateful Memes), we only include non-harmful examples (e.g., Figure 3) in the presentation of this paper. There are other potential societal consequences of our work, none which we feel must be specifically highlighted here.

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

ew.net/forum?id=ekGV1VLuwB.

## A. Point-wise Partial Information Decomposition

In this work, we base our hypotheses on the Point-wise Partial Information Decomposition (PPID) framework (Ince, 2017; Finn & Lizier, 2018; Finn et al., 2018). The objective of decomposing information is to distinguish the contributions by a set of sources. Similar to the seminal work on PID (Williams & Beer, 2010), the contributions are characterized by redundant, unique, and synergistic interactions between the input sources. We provide additional details in this appendix for the formulations in Section 3.

**Point-wise Information Terms**  The point-wise information terms provided by the text ($x_T$) and visual ($x_V$) modalities are defined as $i(x_V; y)$ and $i(x_T; y)$ respectively in this work. In the general form, these terms are expressed, from first principles (Finn & Lizier, 2018), as:

$$i(x; y) = \log \frac{p(y|x)}{p(y)} = \log \frac{p(x, t)}{p(x)p(y)} = \log \frac{p(x|y)}{p(x)} \tag{1}$$

with $x$ being the input source (we contextualize this as either modalities in this work) and $y$ as the target event. Note that the average mutual information $I(X; Y)$ in a distribution would be the expectation of this point-wise information term over all events $\langle i(x; y) \rangle$. Given that the point-wise entropy of an event is $h(x) = -\log p(x)$, we can also expand $i(x; y)$ to arrive at the formulation of the point-wise terms in Section 3.2.1:

$$i(x; y) = \log \frac{p(x|y)}{p(x)} = -\log p(x) - (-\log p(x|y)) = h(x) - h(x|y) \tag{2}$$

This expansion further decomposes the point-wise information terms into two non-negative point-wise entropy terms, $h(x)$ and $h(x|y)$, which serve as the point-wise information specificity and ambiguity respectively (also defined in Section 3.2.1).

We also expand on the point-wise ambiguity term $i^-(x; y) = h(x|y)$ to estimate multimodal interactions in real-world datasets.

$$i^-(x; y) = h(x|y) = -\log p(x|y) \tag{3}$$

Applying Baye's Rule, $p(x|y) = \frac{p(y|x)p(x)}{p(y)}$:

$$i^-(x; y) = -\log \frac{p(y|x)p(x)}{p(y)} \tag{4}$$

Expanding the Logarithm:

$$i^-(x; y) = -\log p(x) + \log p(y) - \log p(y|x)$$
$$i^-(x; y) = h(x) + \log p(y) - \log p(y|x) \tag{5}$$

In this form, we can estimate $h(x)$ through an entropy estimator as described in Section 4.1.2 and detailed in Appendix B. Additionally, we can compute $p(y)$ and $p(y|x)$ with the trained classifiers (also detailed in the same section and appendix). As a result, we generalize equation (5) to form the basis for our formulation in step 7 of Algorithm 1 to estimate multimodal interactions:

$$i^-(X_m; Y) \leftarrow H_\theta(X_m) + \log P(Y) - \log P_\theta(Y|X_m)$$

where each of these terms are vectors of size $N \times 1$ containing the values for each sample in the dataset. Note that $P(Y)$ is simply the probability of the label (e.g. the correct answer or class) for each sample. In the discrete case, the probability of a label $y \in Y$ can be derived from the frequency of $y$ in the entire dataset: $P(y = Y) = \frac{\text{freq(y)}}{N}$

# B. Multimodal Interaction Estimator

We estimate multimodal interactions in real-world datasets by adapting the Lightweight Sample-wise Multimodal Interaction estimator (Yang et al., 2025b) as outlined in Algorithm 3. We utilized SigLIP2 (Tschannen et al., 2025) to extract text and image features of different dimensions (compressed with Principle Component Analysis) and compared the relative change between the meaningful (generated) captions against the random text. The results (Table 5) that meaningful captions consistently yield a substantial increase in redundancy ($R$), with gains reaching as high as **+87.8%** in the unseen test set. The ablation study with random text features produces a much weaker effect, peaking at a **+30.6%** increase. This demonstrates that the increase in redundancy (nearly three times greater) is critically dependent on the semantic relevance of the text.

---

**Algorithm 3** LSMI Estimator $(\mathcal{D}, \mathcal{F})$

---

**Require:** Dataset $\mathcal{D} = \{(x_V, x_T, y)_n\}_{n=1}^N$;
**Require:** Embedding Model $\mathcal{F}(\mathcal{D})$
 1: $X_V, X_T \leftarrow \mathcal{F}(\mathcal{D})$
 2: Train entropy estimators: $H_\theta$
 3: Train unimodal classifiers: $P_\theta(Y|X_V), P_\theta(Y|X_T)$
 4: Train multimodal classifier: $P_\theta(Y|X_V, X_T)$
 5: Compute likelihood of labels: $P(Y)$
 6: Let $X_m$ be $X_V, X_T$, or both $(X_V, X_T)$
 7: $i^+(X_m; Y) \leftarrow H_\theta(X_m)$
 8: $i^-(X_m; Y) \leftarrow H_\theta(X_m) + \log P(Y) - \log P_\theta(Y|X_m)$
 9: $\mathbf{r} = r^+(X_V, X_T; Y) - r^-(X_V, X_T; Y)$
10: $\mathbf{u}_V = i(X_V; Y) - r$
11: $\mathbf{u}_T = i(X_T; Y) - r$
12: $\mathbf{s} = i(X_V, X_T; Y) - r - u_V - u_T$
13: **Output:** $R, U_V, U_T, S \leftarrow \mathbf{E}[r], \mathbf{E}[u_V], \mathbf{E}[u_T], \mathbf{E}[s]$

---

## B.1. LSMI Training

For reproducibility, we report the training parameters for the multimodal interaction estimator. Experiments on interaction estimators were conducted on two RTX A6000 GPUs with a fixed random seed of 42 and the data loading was configured with a batch size of 64 and 8 parallel workers. We optimized both the discriminators and the entropy estimators using the Adam optimizer with a learning rate of $10^{-3}$ with a step learning rate scheduler.

To control for overfitting, we employed an early stopping mechanism monitored on a validation set. Training terminated if the validation loss did not improve by at least $1 \times 10^{-4}$ for 5 consecutive epochs. The classifiers were trained for a maximum of 30 epochs (though the model usually converges in 13-14 epochs), while the entropy estimators were trained for a maximum of 80 epochs. We allocate more epochs to the entropy estimators to help them converge on high-dimensional continuous features.

**Uni-modal and Multimodal Classifiers.** Three distinct discriminator networks were instantiated to estimate the posterior probabilities terms: separate uni-modal classifiers for the image and text streams, and a joint classifier that is trained by concatenating features from both modalities. All discriminators share a consistent architecture comprising of three fully connected layers, forming two hidden layers—with 512 units each—with ReLU activations. These networks are trained simultaneously by minimizing the sum of the cross-entropy losses from all three classification heads.

**Entropy Estimators** The entropy estimator was implemented using KNIFE (Pichler et al., 2022): a Gaussian Mixture Model (GMM) configured with $K = 6$ mixture components. This method parameterizes the covariance structure using a learnable lower-triangular matrix to capture inter-feature dependencies. The estimators for the image and text modalities are instantiated as separate modules and are trained simultaneously; the global objective function is the sum of the negative log-likelihoods of the individual (modality) feature distributions.

*Table 5.* Comparison of MI components before and after captioning, segmented by SigLIP model size and an ablation study using random text features. Each cell shows the absolute values as *Before* → *After*, with the percentage change ($\Delta\%$) in parentheses. The components are redundancy ($R$), unique image information ($U_1$), unique text information ($U_2$), and synergy ($S$). Bold values indicate changes that align with our objective of increasing $R$ while decreasing $U_1$.

| **Smaller Model (`siglip2-base-patch32-256`)** | | | |
|---|---|---|---|
| *Raw Features (768 dimensions)* | | | |
| **Data Split** | $R$ | $U_1$ | $U_2$ | $S$ |

| Data Split | $R$ | $U_1$ | $U_2$ | $S$ |
|---|---|---|---|---|
| Train | $0.091 \rightarrow 0.116$ (**+27.8%**) | $0.189 \rightarrow 0.164$ (**-13.3%**) | $-0.048 \rightarrow 0.000$ (+100.0%) | $0.000 \rightarrow -0.006$ (-) |
| Validation | $0.043 \rightarrow 0.063$ (**+47.9%**) | $0.143 \rightarrow 0.122$ (**-14.4%**) | $0.000 \rightarrow 0.000$ (0.0%) | $0.002 \rightarrow -0.001$ (-135.0%) |
| Test | $0.044 \rightarrow 0.070$ (**+59.8%**) | $0.134 \rightarrow 0.108$ (**-19.5%**) | $-0.002 \rightarrow -0.012$ (-666.7%) | $0.000 \rightarrow 0.000$ (0.0%) |
| *PCA Features (512 dimensions)* | | | | |
| Train | $0.063 \rightarrow 0.161$ (**+156.4%**) | $0.264 \rightarrow 0.164$ (**-37.8%**) | $-0.020 \rightarrow 0.000$ (+100.0%) | $0.000 \rightarrow 0.072$ (-) |
| Validation | $0.043 \rightarrow 0.069$ (**+58.3%**) | $0.137 \rightarrow 0.114$ (**-17.1%**) | $0.000 \rightarrow 0.000$ (+100.0%) | $0.000 \rightarrow -0.010$ (-) |
| Test | $0.046 \rightarrow 0.062$ (**+35.2%**) | $0.137 \rightarrow 0.123$ (**-10.4%**) | $-0.003 \rightarrow 0.000$ (+100.0%) | $0.000 \rightarrow -0.020$ (-) |
| *PCA Features (256 dimensions)* | | | | |
| Train | $0.059 \rightarrow 0.043$ (-27.4%) | $0.335 \rightarrow 0.332$ (-0.9%) | $-0.016 \rightarrow 0.000$ (+100.0%) | $0.000 \rightarrow 0.019$ (-) |
| Validation | $0.046 \rightarrow 0.046$ (**+0.4%**) | $0.146 \rightarrow 0.152$ (+4.1%) | $-0.002 \rightarrow -0.003$ (-8.7%) | $0.000 \rightarrow 0.000$ (0.0%) |
| Test | $0.042 \rightarrow 0.049$ (**+16.3%**) | $0.136 \rightarrow 0.134$ (**-1.1%**) | $0.000 \rightarrow -0.007$ (-) | $0.002 \rightarrow 0.000$ (-100.0%) |

| **Larger Model (`siglip2-giant-opt-patch16-384`)** | | | | |
|---|---|---|---|---|
| *Raw Features (1536 dimensions)* | | | | |
| Train | $0.101 \rightarrow 0.170$ (**+69.2%**) | $0.272 \rightarrow 0.203$ (**-25.5%**) | $-0.058 \rightarrow 0.000$ (+100.0%) | $0.000 \rightarrow 0.029$ (-) |
| Validation | $0.054 \rightarrow 0.065$ (**+20.1%**) | $0.225 \rightarrow 0.215$ (**-4.8%**) | $-0.011 \rightarrow 0.000$ (+100.0%) | $0.000 \rightarrow -0.019$ (-) |
| Test | $0.056 \rightarrow 0.079$ (**+42.1%**) | $0.176 \rightarrow 0.152$ (**-13.4%**) | $-0.014 \rightarrow 0.000$ (+100.0%) | $0.000 \rightarrow -0.024$ (-) |
| *PCA Features (1024 dimensions)* | | | | |
| Train | $0.055 \rightarrow 0.232$ (**+319.3%**) | $0.347 \rightarrow 0.171$ (**-50.6%**) | $-0.013 \rightarrow 0.000$ (+100.0%) | $0.000 \rightarrow 0.086$ (-) |
| Validation | $0.044 \rightarrow 0.093$ (**+108.8%**) | $0.213 \rightarrow 0.164$ (**-22.8%**) | $-0.001 \rightarrow 0.000$ (+100.0%) | $0.000 \rightarrow -0.030$ (-) |
| Test | $0.050 \rightarrow 0.094$ (**+87.8%**) | $0.173 \rightarrow 0.127$ (**-26.9%**) | $-0.008 \rightarrow 0.000$ (+100.0%) | $0.000 \rightarrow -0.033$ (-) |
| *PCA Features (512 dimensions)* | | | | |
| Train | $0.049 \rightarrow 0.226$ (**+360.3%**) | $0.337 \rightarrow 0.158$ (**-53.0%**) | $-0.006 \rightarrow 0.000$ (+100.0%) | $0.000 \rightarrow 0.091$ (-) |
| Validation | $0.043 \rightarrow 0.094$ (**+116.1%**) | $0.213 \rightarrow 0.167$ (**-21.8%**) | $0.000 \rightarrow 0.000$ (+100.0%) | $0.000 \rightarrow -0.026$ (-) |
| Test | $0.053 \rightarrow 0.096$ (**+79.0%**) | $0.172 \rightarrow 0.128$ (**-25.5%**) | $-0.011 \rightarrow 0.000$ (+100.0%) | $0.000 \rightarrow -0.029$ (-) |

| **Ablation with Random Text Features (Larger Model)** | | | | |
|---|---|---|---|---|
| *Raw Features (1536 dimensions) with Random Text* | | | | |
| Train | $0.101 \rightarrow 0.069$ (-31.6%) | $0.272 \rightarrow 0.304$ (+11.6%) | $-0.058 \rightarrow -0.023$ (+60.3%) | $0.000 \rightarrow 0.000$ (0.0%) |
| Validation | $0.054 \rightarrow 0.057$ (**+5.4%**) | $0.225 \rightarrow 0.223$ (**-1.1%**) | $-0.011 \rightarrow -0.014$ (-25.5%) | $0.000 \rightarrow 0.000$ (0.0%) |
| Test | $0.056 \rightarrow 0.065$ (**+15.5%**) | $0.176 \rightarrow 0.167$ (**-5.4%**) | $-0.014 \rightarrow -0.022$ (-60.0%) | $0.000 \rightarrow 0.000$ (0.0%) |
| *PCA Features (1024 dimensions) with Random Text* | | | | |
| Train | $0.055 \rightarrow 0.068$ (**+24.0%**) | $0.347 \rightarrow 0.338$ (**-2.6%**) | $-0.013 \rightarrow 0.000$ (+100.0%) | $0.000 \rightarrow 0.051$ (-) |
| Validation | $0.044 \rightarrow 0.063$ (**+42.7%**) | $0.213 \rightarrow 0.196$ (**-7.8%**) | $-0.001 \rightarrow -0.025$ (-2390.0%) | $0.000 \rightarrow 0.000$ (0.0%) |
| Test | $0.050 \rightarrow 0.065$ (**+30.6%**) | $0.173 \rightarrow 0.157$ (**-9.0%**) | $-0.008 \rightarrow -0.027$ (-236.3%) | $0.000 \rightarrow 0.000$ (0.0%) |
| *PCA Features (512 dimensions) with Random Text* | | | | |
| Train | $0.049 \rightarrow 0.061$ (**+24.1%**) | $0.337 \rightarrow 0.327$ (**-2.8%**) | $-0.006 \rightarrow 0.000$ (+100.0%) | $0.000 \rightarrow 0.045$ (-) |
| Validation | $0.043 \rightarrow 0.062$ (**+44.0%**) | $0.213 \rightarrow 0.196$ (**-7.8%**) | $0.000 \rightarrow -0.022$ (-) | $0.000 \rightarrow 0.000$ (0.0%) |
| Test | $0.053 \rightarrow 0.065$ (**+22.1%**) | $0.172 \rightarrow 0.163$ (**-5.1%**) | $-0.011 \rightarrow -0.023$ (-110.9%) | $0.000 \rightarrow 0.000$ (0.0%) |

## B.2. Bi-directional Interaction Transfers

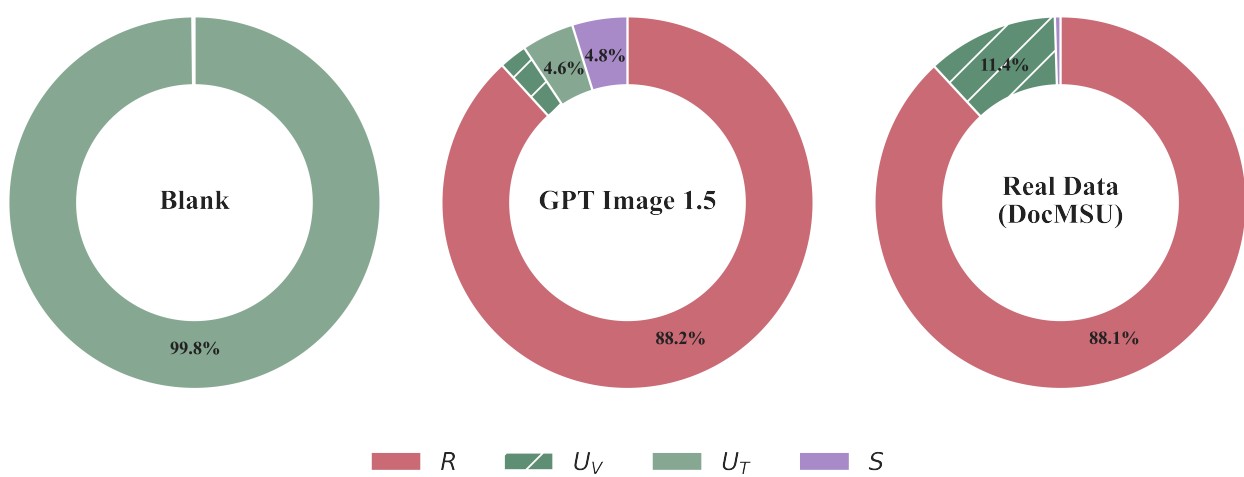

*Figure 7.* Comparison of interactions between synthetically produced (blank) images, (diffusion-based) generated images, and the real data form DocMSU (Du et al., 2024). The generated images successfully converts $U_T$ to $R$ and could even (closely) match the interaction distribution of the real data.

We conducted a separate experiment with a vision-language dataset (1000 samples) for sarcasm detection: DocMSU (Du et al., 2024). First, we compute the interactions of the base dataset and denoted it as Real-Data in Table 6. Next, we curated synthetic blank images (black rectangles) of varying size and dimensions as a baseline. Finally, we generated images with OpenAI's GPT Image 1.5 conditioned on the text modality of the dataset.

The baseline (blank images) expectedly contain nearly no $U_V$ with almost all of the task relevant information in the text (Figure 7 and Table 6). Conversely, the generated images nearly matches the real data for $R$. This implies that generative models could enable a transfer from $U_T$ to $R$ even with zero-shot inferences, providing some preliminary evidence for bi-directional ($U_T \rightarrow R \leftarrow U_V$) transfers.

*Table 6.* Comparison of MI components between generated images and real data (DocMSU dataset). Semantically meaningful images generated by gpt-image-1.5, conditioned on the text modality of each sample, increases redundancy. This serves as preliminary evidence for a text to image interaction transfer: converting $U_2$ to $R$.

| Image Source | $R$ | $U_1$ | $U_2$ | $S$ |
|---|---|---|---|---|
| | Training Set | | | |
| Blank Images | 0.0000 | 0.0000 | 0.6747 | $-0.0001$ |
| gpt-image-1.5 | 0.6732 | 0.0034 | 0.0000 | 0.0152 |
| Real-Data | 0.6736 | 0.0147 | 0.0000 | 0.0039 |
| | Validation Set | | | |
| Blank Images | 0.0000 | 0.0000 | 0.5757 | 0.0032 |
| gpt-image-1.5 | 0.4997 | 0.0089 | 0.0890 | 0.0278 |
| Real-Data | 0.5887 | 0.0766 | 0.0000 | 0.0000 |
| | Test Set | | | |
| Blank Images | 0.0000 | 0.0000 | 0.5259 | $-0.0152$ |
| gpt-image-1.5 | 0.5453 | 0.0347 | 0.0000 | 0.0506 |
| Real-Data | 0.5295 | 0.1409 | 0.0000 | 0.0061 |

# C. Supervised Fine-tuning Vision Language Models

## C.1. SFT Details

We perform Supervised Fine-Tuning (SFT) on both the SmolVLM and LLaVA-OneVision families by building from the SmolVLM GitHub repository. We utilize the established training setting to fine-tune the VLMs. Specifically, we trained the models for 1 epoch on a compute node with 4 H100 GPUs. The global effective batch size is set to 64 (configured with a per-device batch size of 8 and 2 gradient accumulation steps).

In the training, we also adopt a Parameter-Efficient Fine-Tuning (PEFT) strategy with Low-Rank Adaptation (Hu et al., 2022). We froze the vision tower, modality connector, and language backbone, and only update the adapter layers during SFT at a fixed learning rate of $5 \times 10^{-5}$ with a 0.03 warmup ratio. We further conserve GPU memory by using BF16 precision and enabled gradient check-pointing to accommodate a maximum sequence length of 8192 tokens.

Under this configuration, fine-tuning the 8B parameter models required approximately 4 hours, amounting to roughly 16 GPU (H100) hours per run. Training with the same dataset but with the additional captions added to 25% or 50% of the samples did not significantly affect the duration of the SFT.

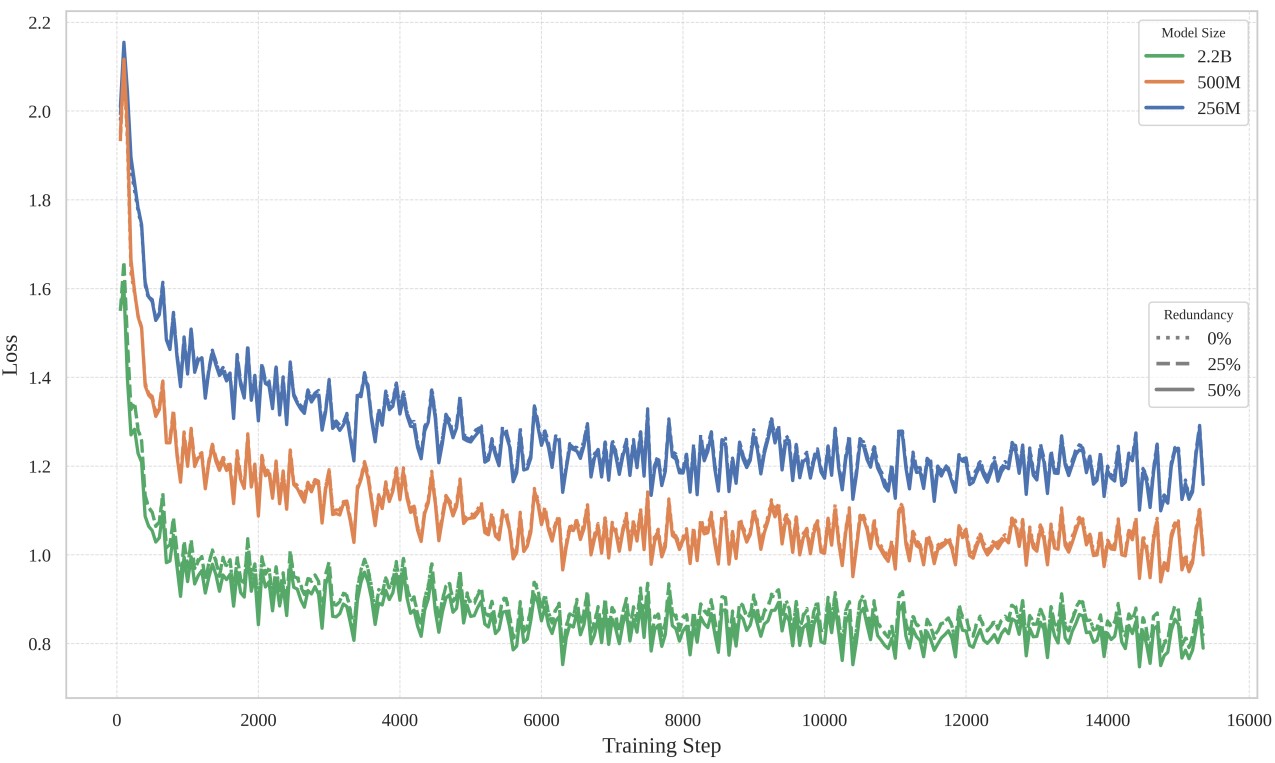

*Figure 8.* The loss curves of training SmolVLM 256M, 500M, and 2B parameter sizes.

## C.2. Instruction Dataset

We curate a balanced training mixture from the Cauldron dataset (Laurençon et al., 2024) comprising of seven categories (Figure 9): *OCR/Docs/Text*, *Captioning*, *Reasoning/Math/Logic*, *Charts/Figures*, *Real-world VQA*, *Tables*, and *Screenshot-to-Code*. The distribution of the datasets is detailed in Table 7.

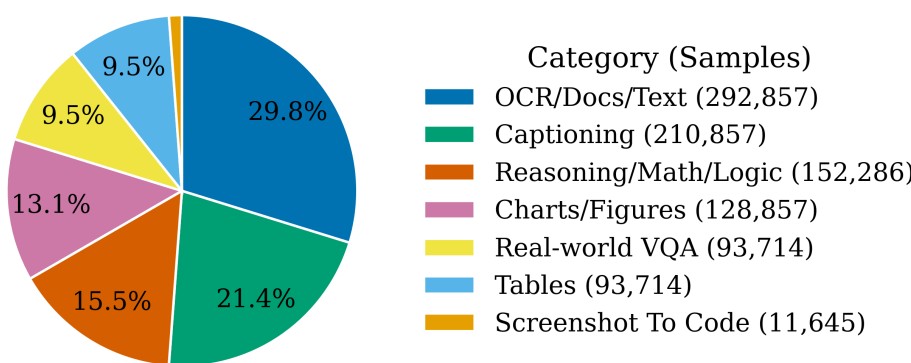

**SFT Mixture Dataset Composition**
**(N = 983,930)**

Category (Samples)
- OCR/Docs/Text (292,857)
- Captioning (210,857)
- Reasoning/Math/Logic (152,286)
- Charts/Figures (128,857)
- Real-world VQA (93,714)
- Tables (93,714)
- Screenshot To Code (11,645)

*Figure 9.* Breakdown of the SFT mixture with the seven categories. There are a total of 983,930 samples in the training data after filtering out samples with excessively large media in the dataset. The distribution of each dataset before filtering is detailed in Table 7.

**Data Preparation.** To ensure diverse representation within each category $\mathcal{C} = \{d_1, d_2, \ldots, d_n\}$, we employ a temperature-based re-weighting strategy to allocate the sampling budget. The weight for a constituent dataset $d_i$ is calculated as $W_i = N_i^\tau$, where $N_i$ denotes the number of available samples and $\tau \in [0, 1]$ is the smoothing temperature. In our implementation, we set $\tau = 0.5$; this square-root scaling effectively boosts the representation of smaller, specialized subsets while still respecting the underlying data scale of larger datasets. The resulting training mixture consists of 984,000 multimodal samples before filtering for quality.

Prior to training, we process the raw Cauldron subsets to ensure data quality and compatibility. We excluded the `clevr_math` and `okvqa` subsets to avoid data duplication and external dependency issues, respectively. The remaining images are standardized to optimize resource usage: images exceeding 1536 pixels on the longest side or a total area of 2 million pixels are downscaled. Additionally, to ensure stability during training, we filter out any samples where the tokenized sequence length exceeds the model's maximum context window of 8192 tokens. This quality control brought the original 984,000 samples down to 983,930 samples (70 samples, most of which from `datikz`, were filtered out).

**Redundancy Injection and Allocation.** To isolate redundant information, we modify predetermined samples with synthetic captions generated by a vision-language model (Qwen2.5-VL-32B-Instruct). These captions are injected into the context of selected samples based on a deterministic hashing strategy. Specifically, we utilize SHA-256 hashing on unique sample identifiers to consistently assign each data point to a specific redundancy tier (0%, 25%, or 50%). The 50% category would contain the samples that were captioned in the 25% variant to ensure progressive scaling. This would ensure that our experiments are reproducible while having a "random" selection of samples being captioned.

*Table 7.* Distribution of Samples in the Cauldron Dataset

| Dataset | Count | Percentage |
|---|---|---|
| cauldron:ai2d:zero | 4,163 | 0.42% |
| cauldron:aokvqa:zero | 6,023 | 0.61% |
| cauldron:chart2text:zero | 2,726 | 0.28% |
| cauldron:chartqa:zero | 2,638 | 0.27% |
| cauldron:clevr:zero | 38,586 | 3.92% |
| cauldron:cocoqa:zero | 20,054 | 2.04% |
| cauldron:datikz:zero | 8,051 | 0.82% |
| cauldron:diagram_image_to_text:zero | 299 | 0.03% |
| cauldron:docvqa:zero | 39,145 | 3.98% |
| cauldron:dvqa:zero | 23,921 | 2.43% |
| cauldron:figureqa:zero | 18,073 | 1.84% |
| cauldron:finqa:zero | 5,367 | 0.55% |
| cauldron:geomverse:zero | 4,456 | 0.45% |
| cauldron:hateful_memes:zero | 4,252 | 0.43% |
| cauldron:hitab:zero | 5,988 | 0.61% |
| cauldron:iam:zero | 5,663 | 0.58% |
| cauldron:iconqa:zero | 7,966 | 0.81% |
| cauldron:infographic_vqa:zero | 10,074 | 1.02% |
| cauldron:intergps:zero | 1,760 | 0.18% |
| cauldron:localized_narratives:zero | 163,198 | 16.59% |
| cauldron:mapqa:zero | 10,909 | 1.11% |
| cauldron:mimic_cgd:zero | 24,566 | 2.50% |
| cauldron:multihiertt:zero | 12,065 | 1.23% |
| cauldron:nlvr2:zero | 19,168 | 1.95% |
| cauldron:ocrvqa:zero | 157,991 | 16.06% |
| cauldron:plotqa:zero | 70,590 | 7.17% |
| cauldron:raven:zero | 11,583 | 1.18% |
| cauldron:rendered_text:zero | 10,000 | 1.02% |
| cauldron:robut_sqa:zero | 12,543 | 1.27% |
| cauldron:robut_wikisql:zero | 19,930 | 2.03% |
| cauldron:robut_wtq:zero | 14,254 | 1.45% |
| cauldron:scienceqa:zero | 3,616 | 0.37% |
| cauldron:screen2words:zero | 15,743 | 1.60% |
| cauldron:spot_the_diff:zero | 6,365 | 0.65% |
| cauldron:st_vqa:zero | 23,104 | 2.35% |
| cauldron:tabmwp:zero | 10,299 | 1.05% |
| cauldron:tallyqa:zero | 19,782 | 2.01% |
| cauldron:tat_qa:zero | 7,803 | 0.79% |
| cauldron:textcaps:zero | 21,947 | 2.23% |
| cauldron:textvqa:zero | 34,593 | 3.52% |
| cauldron:tqa:zero | 5,465 | 0.56% |
| cauldron:vistext:zero | 9,969 | 1.01% |
| cauldron:visual7w:zero | 18,884 | 1.92% |
| cauldron:visualmrc:zero | 11,988 | 1.22% |
| cauldron:vqarad:zero | 1,793 | 0.18% |
| cauldron:vqav2:zero | 49,629 | 5.04% |
| cauldron:vsr:zero | 3,354 | 0.34% |
| cauldron:websight:zero | 3,664 | 0.37% |
| **Total Samples** | **984,000** | **100.00%** |

# D. Experiments in Robustness

In this work, we evaluate VLM robustness by its resilience to ambiguous or corrupted modalities. To this end, we conduct extensive experiments by building on previous benchmarks and frameworks to test for our hypotheses surrounding redundant interactions from Section 3.

## D.1. Ambiguity Experiments

We conducted experiments with manipulated modalities to evaluate robustness against ambiguity. These experiments are built on top of HallusionBench (Guan et al., 2024) which diagnoses hallucinations based on the type of error made by the vision language model. In general, this experiment is conducted by either swapping out the image or the text (questions) to make one modality more ambiguous or misleading such that the model makes an error in its response. The absolute values for each model are detailed in Table 8. Notably, the models trained with additional redundancies have higher consistency and largely lower visual induced errors compared to the baseline models.

Specifically, we characterize the errors into Language Induced (LI) and Visual Induced (VI) hallucinations for better clarity in this work. This would also help us to better attribute mistakes to specific modalities and answer the outlined hypotheses in Sections 3 and 4. The original work defined LI and VI as Language Hallucinations and Visual Illusions.

*Table 8.* Diagnosis of failure modes using HallusionBench (Guan et al., 2024) with counts of Language Induced (LI), Visual Induced (VI), and Mixed errors. **Bold values** indicate best results for each category.

| Model | +R | LI ↓ | VI ↓ | Mix ↓ | Consistency ↑ |
|---|---|---|---|---|---|
| | 0% | 55 | 524 | 175 | 0.2697 |
| SmolVLM 256M | 25% | 64 | 381 | 281 | 0.2943 |
| | 50% | 74 | **371** | 278 | **0.3094** |
| | 0% | 96 | 365 | 278 | 0.3385 |
| SmolVLM 500M | 25% | 94 | 352 | 281 | 0.3342 |
| | 50% | 96 | **254** | 342 | **0.3823** |
| | 0% | 64 | 349 | 275 | 0.3491 |
| SmolVLM 2B | 25% | 73 | 210 | 355 | 0.4110 |
| | 50% | 71 | **156** | 404 | **0.4288** |
| | 0% | 90 | 317 | 273 | 0.3945 |
| LLaVa-OneVision 4B | 25% | 92 | **185** | 361 | 0.4394 |
| | 50% | 82 | 200 | 349 | **0.4399** |
| | 0% | 86 | 218 | 308 | 0.4559 |
| LLaVa-OneVision 8B | 25% | 89 | **159** | 351 | **0.4609** |
| | 50% | 82 | 270 | 252 | 0.4539 |

**Language Induced Hallucinations.** We associate these errors with cases where the model ignores visual information in favor of its parametric knowledge. Following the decision tree methodology in (Guan et al., 2024), we classify an error as Language Induced in two specific scenarios within the Visual Supplement (VS) category—a set of samples that do not necessarily require the image (though it could help) to obtain the correct answer. If the model provides an incorrect answer when no visual input is provided, the error is purely hallucinated from text. More importantly, if the model answers correctly without an image but fails to update its answer when presented with an ambiguous or manipulated image (providing the *same* incorrect response), we also attribute this to LI, a reliance on language priors over visual context.

**Visual Induced Hallucinations.** We associate these errors with a direct misinterpretation of visual features. In our evaluation, an error is classified as Visual Induced if the model fails to answer correctly on the reference control image in the Visual Dependent (VD) category—a set of samples that do require information from the image. Additionally, for both VS and VD categories, if the model answers the control case correctly but provides an incorrect answer (or an "Uncertain" prediction) on the ambiguous (manipulated) test image, we attribute the failure to visual induced hallucinations—the model

*Table 9.* Detailed breakdown of percentage changes per model variant relative to the $0\%$ baseline. $\Delta Acc$: Absolute percentage point increase (e.g., $(Acc_{new} - Acc_{old})/Total$). $\Delta$ **Errors/Consistency**: Relative percentage change (e.g., $(Val_{new} - Val_{old})/Val_{old}$). The **Avg (Macro)** rows represent the final values reported in the summary table (Table 3).

| Model | Rate | $\Delta Acc \uparrow$ | $\Delta LI \downarrow$ | $\Delta VI \downarrow$ | $\Delta Mix \downarrow$ | $\Delta Consist. \uparrow$ |
|---|---|---|---|---|---|---|
| *SmolVLM Family* | | | | | | |
| SmolVLM 256M | 25% | +2.48% | +16.36% | -27.29% | +60.57% | +9.12% |
| SmolVLM 500M | 25% | +1.06% | -2.08% | -3.56% | +1.08% | -1.27% |
| SmolVLM 2B | 25% | +4.43% | +14.06% | -39.83% | +29.09% | +17.73% |
| **Avg (Macro)** | **25%** | **+2.65%** | **+9.45%** | **-23.56%** | **+30.25%** | **+8.52%** |
| SmolVLM 256M | 50% | +2.75% | +34.55% | -29.20% | +58.86% | +14.72% |
| SmolVLM 500M | 50% | +4.16% | +0.00% | -30.41% | +23.02% | +12.94% |
| SmolVLM 2B | 50% | +5.05% | +10.94% | -55.30% | +46.91% | +22.83% |
| **Avg (Macro)** | **50%** | **+4.01%** | **+15.16%** | **-38.30%** | **+42.93%** | **+16.82%** |
| *LLaVA-OneVision Family* | | | | | | |
| LLaVA-OV 4B | 25% | +3.72% | +2.22% | -41.64% | +32.23% | +11.38% |
| LLaVA-OV 8B | 25% | +1.15% | +3.49% | -27.06% | +13.96% | +1.10% |
| **Avg (Macro)** | **25%** | **+2.44%** | **+2.86%** | **-34.35%** | **+23.10%** | **+6.23%** |
| LLaVA-OV 4B | 50% | +4.34% | -8.89% | -36.91% | +27.84% | +11.51% |
| LLaVA-OV 8B | 50% | +0.71% | -4.65% | +23.85% | -18.18% | -0.44% |
| **Avg (Macro)** | **50%** | **+2.52%** | **-6.77%** | **-6.53%** | **+4.83%** | **+5.53%** |

attempted to process the new visual information but did so incorrectly.

**Consistency.** Beyond individual hallucination types, we evaluate the consistency of the model. In the evaluation script, the questions are grouped with a unique image. For each image, we compute the total number of questions $t$ and the number of correct responses $c$. The model is considered **consistent on a figure only if it correctly answers every question associated with it**. If the model answers some questions correctly but fails others for the same image, it is flagged as inconsistent, indicating that the correct answers may stem from random guessing rather than robust visual reasoning.

We adopt the macro-average percentage change to **isolate the effects of redundant information** in each VLM family. Because baseline performance varies across scales (e.g., 256M vs. 2B), aggregating raw error counts would allow smaller, less capable models to disproportionately influence the findings. Instead, by measuring the relative shift for each variant first, we can compute the average change in each redundancy level (Table 9) to capture the impact of increasing redundancy on the model family's behavior, independent of individual model performance. The final average macro values (rounded to one decimal place) are reported in the main body of work (Table 3).

## D.2. Corruption Experiments

We conducted experiments with noisy modalities to evaluate robustness against corrupted modalities. As defined in Section 5.3, we evaluate with performance stability $\Delta P$ which measures the degree of change in a model's performance when the modalities are corrupted relative to the untouched modalities. We base our experiments on the GQA (Hudson & Manning, 2019) benchmark which contains the image and text samples that we progressively corrupt by building on the adarobustness GitHub (Chen et al., 2023).

**Image Corruption.** The images are corrupted with Impulse, Gaussian and Shot noise with increasing levels of severity (from one to five). The corrupted performance is computed by keeping the text modality unchanged and using the same image but corrupted at different levels of severity. The clean performance is computed using modalities untouched by any form of corruption. Figure 10 below illustrates the image corruption methods.

**Text Corruption** The text is corrupted with random character insertions, deletions, and replacements with increasing levels of severity (from one to five). The corrupted performance is computed by keeping the image modality unchanged and using the same text but corrupted at different levels of severity. The clean performance is computed using modalities untouched by any form of corruption. Table 10 below illustrates the text corruption methods.

In addition, we follow the conventional practice of ensuring fidelity in the corrupted sentence. Specifically, we set the baseline cosine similarity score between the original sentence and the corrupted sentence to 0.2; this score is computed by a sentence transformer model as set by the original work (Chen et al., 2023). If the two sentences do not meet this similarity score, the corruption method is repeated on the original sentence up to 100 times to maintain some fidelity. We exclude samples that failed to maintain the 0.2 similarity score from the test set.

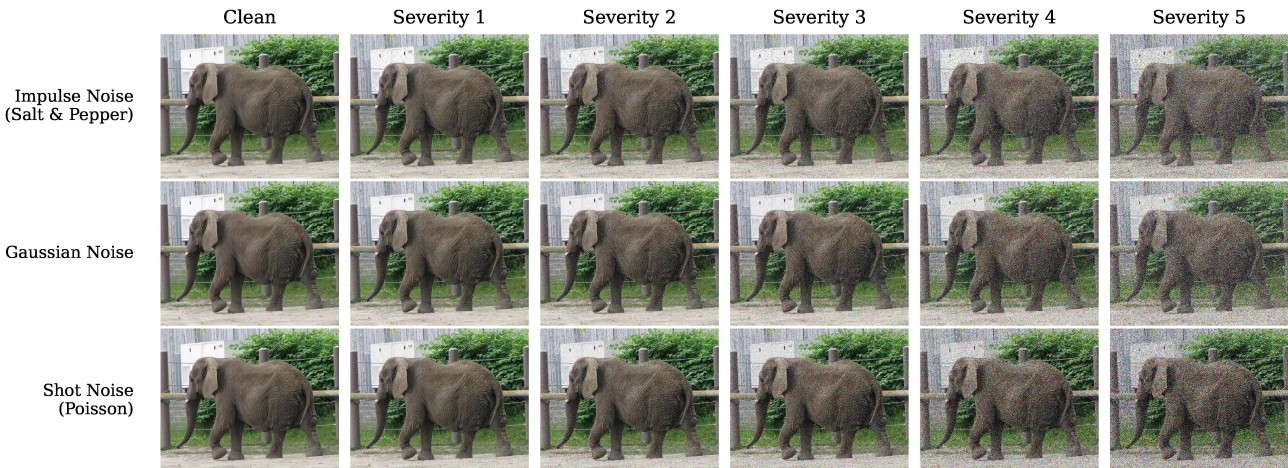

*Figure 10.* An image corruption example from GQA (Hudson & Manning, 2019) with the Impulse, Gaussian, and Shot noise methods from the clean image to the different severity levels of corruption.

*Table 10.* A text corruption method from GQA (Hudson & Manning, 2019) using random character deletion, insertion, and replacement. Naturally, the model suffers the worst performance degradation at severity level 5.

| *Clean* | The elephant is in front of what?    Answer:  A Fence | | |
|---------|------------------------|------------------------|------------------------|
| **Severity** | **Deletion** | **Insertion** | **Replacement** |
| Level 1 | The elephnt is in ront of what? | The eqlephant is in front' of whatS? | The elephant ll in front ot wh:h? |
| Level 3 | Theeephai in ontof what? | The elephQant isM in front of wIhats? | T%e 8lephXnt is inafront ofBohYt? |
| Level 5 | The elephan i nron oh? | The' el;ephGant is i*n f^r9oGnt] o$f1 what? | The \\*eS]aki.is {n Aro3[ op |

*Table 11.* Performance Change $\Delta P$ of SmolVLM (top) and LLaVa-OneVision (bottom) under **Image** and **Text** Corruptions. Values are reported as Mean $\pm$ SD. $\Delta P \approx 0$ indicates robustness; negative values indicate degradation. **Bold** values indicate the most stable configuration (highest mean) for that specific model and severity level.

| Severity | SmolVLM 256M | | | SmolVLM 500M | | | SmolVLM 2.2B | | |
| --- | --- | --- | --- | --- | --- | --- | --- | --- | --- |
| | **0%** | **25%** | **50%** | **0%** | **25%** | **50%** | **0%** | **25%** | **50%** |
| *Image Corruptions* | | | | | | | | | |
| Level 1 | $-2.9 \pm 0.2$ | **+1.0** $\pm 0.2$ | $+0.5 \pm 0.1$ | **-1.6** $\pm 0.2$ | $-2.2 \pm 0.2$ | $-9.3 \pm 0.6$ | $-0.7 \pm 0.7$ | $-0.3 \pm 0.8$ | **+1.6** $\pm 0.2$ |
| Level 2 | $-2.8 \pm 0.1$ | **+0.3** $\pm 0.1$ | $-1.0 \pm 0.6$ | **-2.4** $\pm 0.2$ | $-3.0 \pm 0.1$ | $-8.2 \pm 0.7$ | $-2.2 \pm 0.7$ | $-1.7 \pm 0.6$ | **+0.4** $\pm 0.7$ |
| Level 3 | $-3.1 \pm 0.6$ | **-0.6** $\pm 0.1$ | $-1.6 \pm 0.2$ | $-4.3 \pm 0.1$ | **-3.6** $\pm 0.2$ | $-8.0 \pm 0.9$ | $-3.8 \pm 0.4$ | $-3.8 \pm 0.4$ | **-1.1** $\pm 0.2$ |
| Level 4 | $-6.1 \pm 0.6$ | **-3.9** $\pm 0.8$ | $-4.2 \pm 0.2$ | $-6.7 \pm 0.4$ | **-6.0** $\pm 0.2$ | $-9.0 \pm 1.4$ | $-6.6 \pm 0.3$ | $-7.5 \pm 0.4$ | **-4.0** $\pm 0.4$ |
| Level 5 | $-10.0 \pm 0.8$ | **-5.9** $\pm 0.1$ | $-7.1 \pm 0.4$ | $-11.0 \pm 0.2$ | **-9.8** $\pm 0.6$ | $-11.2 \pm 1.1$ | $-9.7 \pm 0.6$ | $-11.1 \pm 0.4$ | **-7.0** $\pm 0.2$ |
| **Overall** | $-5.0$ | **-1.8** | $-2.7$ | $-5.2$ | **-4.9** | $-9.1$ | $-4.6$ | $-4.9$ | **-2.0** |
| *Text Corruptions* | | | | | | | | | |
| Level 1 | $-28.6 \pm 4.6$ | **-25.5** $\pm 3.9$ | $-26.2 \pm 3.9$ | $-19.7 \pm 3.8$ | **-19.3** $\pm 3.3$ | $-22.6 \pm 3.4$ | **-15.1** $\pm 3.2$ | $-16.9 \pm 3.8$ | $-18.2 \pm 4.3$ |
| Level 2 | $-47.8 \pm 5.9$ | **-40.8** $\pm 5.5$ | $-41.4 \pm 5.1$ | $-32.4 \pm 4.3$ | **-30.3** $\pm 4.4$ | $-31.8 \pm 3.7$ | **-28.8** $\pm 5.2$ | $-30.0 \pm 4.9$ | $-30.4 \pm 5.9$ |
| Level 3 | $-57.6 \pm 6.7$ | **-51.3** $\pm 6.7$ | $-51.8 \pm 6.7$ | $-41.3 \pm 5.8$ | $-39.1 \pm 5.5$ | **-37.6** $\pm 4.1$ | **-38.8** $\pm 7.8$ | $-39.3 \pm 6.8$ | $-41.3 \pm 6.7$ |
| Level 4 | $-64.2 \pm 6.6$ | **-58.3** $\pm 6.4$ | $-58.5 \pm 6.1$ | $-49.5 \pm 7.7$ | $-45.0 \pm 6.9$ | **-42.7** $\pm 7.5$ | $-47.3 \pm 7.6$ | **-47.0** $\pm 7.4$ | $-49.3 \pm 7.5$ |
| Level 5 | $-69.3 \pm 7.0$ | $-63.8 \pm 7.0$ | **-63.5** $\pm 6.9$ | $-57.1 \pm 9.6$ | $-50.6 \pm 8.9$ | **-47.8** $\pm 9.9$ | $-54.9 \pm 9.6$ | **-53.1** $\pm 8.0$ | $-55.3 \pm 8.5$ |
| **Overall** | $-53.5$ | **-47.9** | $-48.3$ | $-40.0$ | $-36.9$ | **-36.5** | **-37.0** | $-37.3$ | $-38.9$ |
| $P_{Clean}$ | 29.8 | 31.7 | 31.1 | 31.7 | 31.3 | 28.9 | 39.7 | 39.6 | 39.7 |

| Severity | LLaVa-OneVision 4B | | | LLaVa-OneVision 8B | | |
| --- | --- | --- | --- | --- | --- | --- |
| | **0%** | **25%** | **50%** | **0%** | **25%** | **50%** |
| *Image Corruptions* | | | | | | |
| Level 1 | $+0.1 \pm 0.3$ | **+1.0** $\pm 0.2$ | $+0.5 \pm 0.6$ | **-0.9** $\pm 0.2$ | $-1.0 \pm 0.4$ | $-1.1 \pm 0.8$ |
| Level 2 | $-0.6 \pm 0.0$ | **+2.8** $\pm 0.4$ | $+0.7 \pm 0.3$ | $-2.0 \pm 0.1$ | **-1.5** $\pm 0.7$ | $-1.9 \pm 0.2$ |
| Level 3 | $+0.3 \pm 0.3$ | **+5.0** $\pm 1.2$ | $+1.1 \pm 0.1$ | $-2.8 \pm 0.2$ | **-2.3** $\pm 0.3$ | $-2.7 \pm 0.4$ |
| Level 4 | $+0.1 \pm 0.4$ | **+7.4** $\pm 0.6$ | $+0.1 \pm 0.3$ | $-3.6 \pm 0.2$ | $-3.1 \pm 0.1$ | **-3.0** $\pm 0.4$ |
| Level 5 | $-0.7 \pm 0.4$ | **+9.2** $\pm 0.3$ | $-1.1 \pm 0.2$ | $-5.5 \pm 0.4$ | **-4.2** $\pm 0.1$ | $-5.1 \pm 0.4$ |
| **Overall** | $-0.1$ | **+5.1** | $+0.3$ | $-3.0$ | **-2.4** | $-2.8$ |
| *Text Corruptions* | | | | | | |
| Level 1 | $-25.0 \pm 5.9$ | $-34.8 \pm 7.1$ | **-24.2** $\pm 6.3$ | $-23.2 \pm 6.0$ | $-25.3 \pm 7.0$ | **-21.9** $\pm 5.7$ |
| Level 2 | $-43.8 \pm 9.4$ | $-56.4 \pm 10.4$ | **-43.4** $\pm 10.4$ | $-43.6 \pm 10.3$ | $-48.0 \pm 11.8$ | **-42.3** $\pm 11.0$ |
| Level 3 | $-57.7 \pm 10.9$ | $-71.2 \pm 10.3$ | **-56.7** $\pm 11.7$ | $-59.3 \pm 12.1$ | $-64.3 \pm 12.1$ | **-59.0** $\pm 13.5$ |
| Level 4 | **-66.6** $\pm 10.7$ | $-80.2 \pm 9.2$ | $-67.0 \pm 11.1$ | $-70.3 \pm 11.4$ | $-74.8 \pm 10.7$ | **-69.9** $\pm 12.5$ |
| Level 5 | $-74.6 \pm 9.4$ | $-86.2 \pm 6.9$ | **-73.6** $\pm 9.5$ | **-77.9** $\pm 9.5$ | $-82.3 \pm 8.5$ | $-78.2 \pm 10.4$ |
| **Overall** | $-53.5$ | $-65.7$ | **-53.0** | $-54.8$ | $-58.9$ | **-54.3** |
| $P_{Clean}$ | 33.3 | 28.6 | 35.8 | 37.2 | 37.7 | 38.3 |

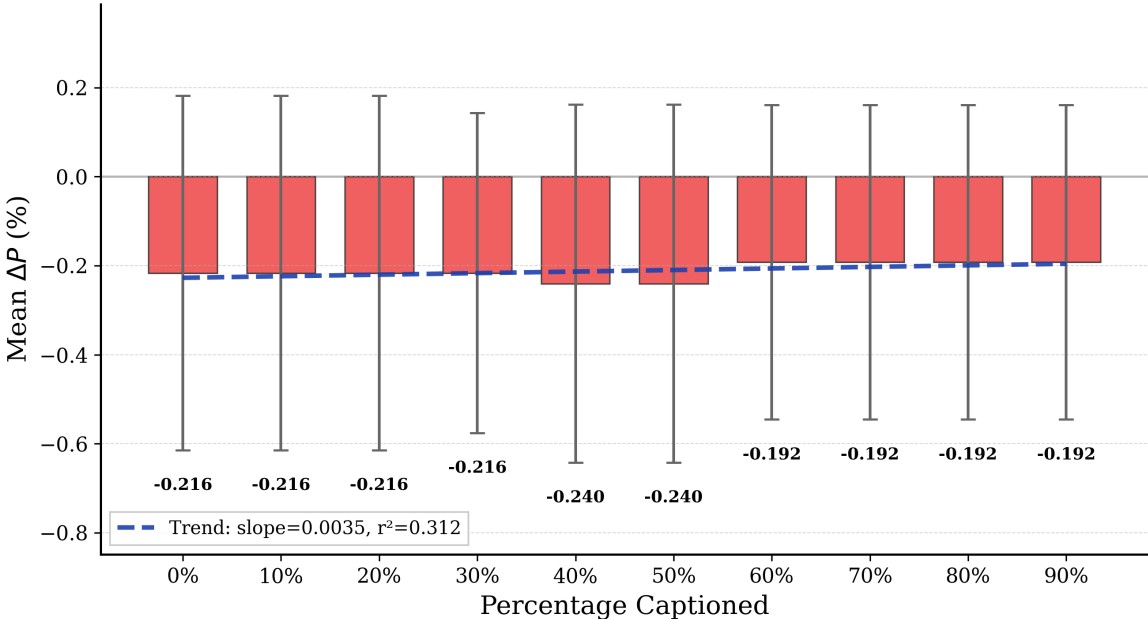

*Figure 11.* Trend of average $\Delta P$ across all five severity levels of impulse image corruption as the threshold of samples captioned by the MI GATE increase. This task involves detecting hate speech with the cauldron variant of the Hateful Memes (Kiela et al., 2020) dataset. Notably, there is a slight positive trend in increasing redundant interactions with lower standard deviations by captioning more samples.

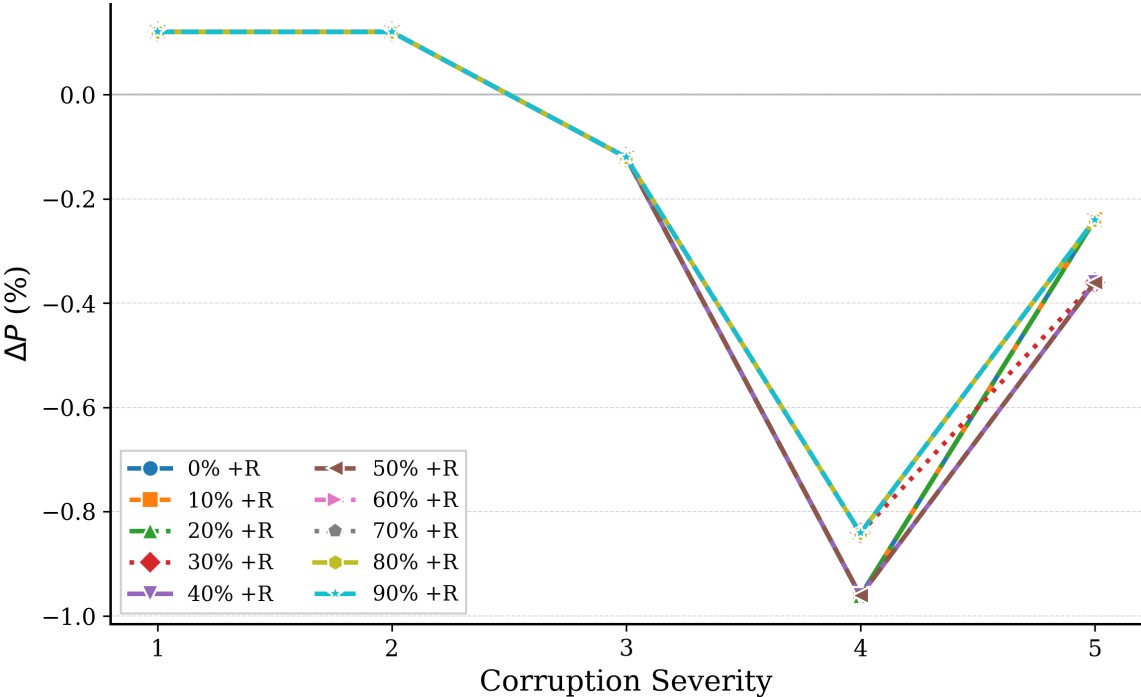

*Figure 12.* Average performance stability $\Delta P$ of each SmolVLM at specific levels of corruption severity for the Cauldron variant of the Hateful Memes dataset (Kiela et al., 2020). The SmolVLM are trained at 0-90% of the samples captioned by Qwen2.5-VL-32B-Instruct.

