# OpenReview forum: "Self-Captioning Multimodal Interaction Tuning: Amplifying Exploitable Redundancies for Robust Vision Language Models"
_ICML.cc/2026/Conference — ICML 2026 regular_

### Official Review · Reviewer_T4Fh · 2026-03-11

**Soundness:** 3
**Presentation:** 2
**Significance:** 3
**Originality:** 4
**Overall Recommendation:** 5
**Confidence:** 4

**Summary:**

The paper proposes a Self-Captioning Multimodal Interaction Tuning (MIT) workflow to address hallucinations and robustness issues in Vision Language Models (VLMs) caused by ambiguous or corrupted inputs. The Multimodal Interaction (MI) Gate decomposes data attributes into redundant (shared), unique (exclusive), and synergistic (emergent) information, and specifically identifies data points with high levels of unique visual information. Based on that, they transfer unique visual cues into redundant interactions through image captioning. Experimental results demonstrate that amplifying these redundancies significantly enhances model resilience against external noise, reducing visual-induced errors by up to 38.3% and improving response consistency by 16.8%.

**Compliance With Llm Reviewing Policy:**

Affirmed.

**Final Justification:**

The authors have fully addressed my concerns. I believe this paper will be useful for training small models or for adapting to new domains such as the medical field. The proposed MI Gate and self-captioning workflow offer a systematic approach to enhancing model robustness. Therefore, I am increasing my score to 5.

**Key Questions For Authors:**

There are several unclear points regarding the following:
1. MI Gate implementation: Ambiguity remains regarding the training sequence and the exact number of required estimators.
2. Performance along different captioners: It is unclear how scaling the quality of the captioning model affects redundancy levels.
3. Data analysis: Certain data interpretations, such as negative values, lack detailed explanation or clear references.

Please refer to the Weaknesses section for more details.

**Limitations:**

Yes

**Strengths And Weaknesses:**

## Strengths
- Rather than simply improving performance through captioning, the paper offers an insightful analysis of how systematically categorized data attributes affect model learning. The potential expansion to other domains like audio is highly promising.
- The pointwise information decomposition framework is well-designed and theoretically grounded.
- The authors conducted extensive experiments across various tasks and model scales to support their hypotheses.

## Weaknesses (with Questions)
- The paper is somewhat dense; for instance, the term $h(\cdot)$ (entropy) is introduced without an explicit definition for readers unfamiliar with information theory.
- The experiments use relatively older models (SmolVLM, LLaVA-1.5). It would be valuable to see if these findings hold for the latest  models like Qwen3-VL or InternVL3.5.
- Does the model-agnostic nature of the MI Gate mean the same selection of samples can be used for any VLM on the same task?
- Regarding Algorithm 1, how is the entropy estimator trained before the classifier, and are exactly three separate estimators (vision, text, multimodal) required?
- Why did you choose the "self" captioning setup? Would using a superior captioner lead to even higher redundancy, and conversely, how much does a weaker captioner degrade the quality of the "Interaction Transfer"?
- In Table 1, what is the physical interpretation of $U_T$ having a negative value?
- Including Synergy ($S$) values in Table 1 would help clarify how the analysis applies to general datasets that may not have high synergy.
- The claims in lines 424-425 lack a clear pointer for the results.

---

> ### Author Rebuttal · Authors · 2026-03-28
>
> We thank Reviewer T4Fh for the encouraging feedback and for recognizing the analysis and extensive experiments supporting our framework. We address the specific questions below:
>
> ### W1: Paper density and undefined terms
> We will add a brief accessible primer on key concepts, specifically entropy, mutual information, and their relationship to the PID framework,  at the beginning of Section 3 in the revision. We will also include intuitive examples of redundant, unique, and synergistic interactions to further lower the conceptual barrier. We note that Sec 3.1 already provides formal definitions of the core interaction terms; the revision will complement these with informal explanations to make the paper more accessible to readers unfamiliar with information theory.
>
> ### W2: Validation on recent models
> Our primary aim is to isolate the effects of adjusting multimodal interactions, requiring models that are open-source in both weights and pretraining/fine-tuning data. Models like Qwen3-VL and InternVL 3.5 do not disclose their pretraining data, introducing confounding factors that make it difficult to attribute observed changes solely to redundancy. That said, our findings are consistent across two model families and five model sizes (256M–8B), providing strong evidence of generalizability across architectures and scales. We acknowledge that extending to more recent models is a worthwhile direction and leave this to future work.
>
> ### W3: Model-agnostic nature of the MI Gate
> Yes, the same augmented dataset can in principle be used to fine-tune any VLM on the same task. The MI Gate operates entirely on the dataset rather than on any specific (VL) model. We note a soft dependency on the choice of embedding model in Algorithm 1, which could marginally influence which samples are selected; however, the core methods operate on the data. We will clarify this explicitly in the revision.
>
> ### W4: Algorithm 1 training sequence and number of estimators
> We clarify the training sequence and estimator requirements in Algorithm 1:
>
> Training sequence: The entropy estimator (KNIFE, Pichler et al., 2022) is trained first on the embedded representations (line 1) independently of the classifiers, as it only requires the marginal distributions of the modalities. The classifiers are subsequently trained using the embedded representations to estimate the conditional distributions $P(Y|X_V)$ and $P(Y|X_T)$.
>
> Number of estimators: Three separate classifiers are required — one for each of the vision-only, text-only, and multimodal settings — to estimate $i(X_V; Y)$, $i(X_T; Y)$, and $i(X_V, X_T; Y)$ respectively. These 3-layer MLPs, making the additional computational. The entropy estimator is shared across modalities.
>
> We acknowledge that Algorithm 1 could be presented more explicitly and will add a dedicated clarification in the revision.
>
> ### W5: Self-captioning setup and captioner quality
> The self-captioning setup was chosen for reproducibility and to avoid introducing external model dependencies; using the same model family for both captioning and fine-tuning eliminates additional confounding factors. That said, the effect of captioner quality is addressed in Sec 5.1, where we compare SmolVLM-2B and Qwen2.5-VL-32B as captioners. The results show that larger captioners produce stronger interaction transfers, suggesting that a superior captioner would indeed yield higher redundancy. Also, Hypothesis 4 (Section 4.1.3) addresses the effect of weaker captioners: captioning a larger proportion of samples averages out noise from erroneous captions, mitigating quality degradation. This is supported empirically; even SmolVLM-2B, a relatively small captioning model, consistently improves robustness (Figure 3). We will add a clearer pointer to Table 1 in the revision to make this analysis more discoverable as well as the suggested inclusion of synergistic values.
>
> ### W6: Physical interpretation of negative values in Table 1
> Negative values in point-wise information decomposition are theoretically valid and well-documented in the PPID literature (Finn & Lizier, 2018). They arise from the decomposition of point-wise information into specificity $i^+$ and ambiguity $i^-$ (Sec 3.2.1). A negative value indicates that a modality is misinformative about the task for that sample; i.e., it increases surprise about y rather than reducing it. Concretely, we use these terms to compute the $U_T$ values. The negative $U_T$ values observed can be interpreted as  the random text additions confusing the model. Table 1 directly illustrates this: meaningless character sequences added to the text modality actively confused the model's association between the text and the task label, increasing rather than reducing uncertainty. This is consistent with our finding that semantic content is the primary driver of interaction transfer (Sec 5.1). We will add an explicit clarification of negative values in the revision to make this interpretation more accessible.

---

> > ### Author Rebuttal · Reviewer_T4Fh · 2026-04-03
> >
> > I appreciate the authors' clarification regarding the choice of models. I fully understand that models with transparent training data were used to ensure scientific rigor and to prevent data contamination issues. The reason I mentioned experiments on the latest models was that I wanted to verify whether this methodology has the 'practical scalability' to be immediately applicable to the latest high-performance VLMs.
> >
> > Nevertheless, I believe this paper will be useful for training small models or training in new domains such as the medical field. Therefore, I am increasing my score to 5.

---

> > > ### Author Response · Authors · 2026-04-04
> > >
> > > We sincerely thank reviewer T4Fh for raising their scores and for their encouraging comments. We agree that applying this framework to specialized domains (e.g., medical fields) would be a promising and exciting direction for future research. We appreciate your valuable time and insightful feedback throughout this review process.

---

### Official Review · Reviewer_ms6p · 2026-03-12

**Soundness:** 1
**Presentation:** 3
**Significance:** 2
**Originality:** 2
**Overall Recommendation:** 3
**Confidence:** 3

**Summary:**

This paper proposes a multimodal interaction tuning method with a self-captioning method, aiming at improving the robustness of vision language models for ambiguous or corrupted input signals. Concretely, it introduces a module named multimodal interaction gate, which converts visual information to textual descriptions to enhance the redundancy between visual and textual modalities.Through theoretical analysis and extensive experiments, this paper validates the effectiveness of its proposed method across multiple tasks.

**Compliance With Llm Reviewing Policy:**

Affirmed.

**Final Justification:**

Considering all the reviewers, I keep my rating.

**Key Questions For Authors:**

1. From the table 4, the performance on the TextVQA dataset is not satisfying, which needs further explainations.

2. The proposed method only implements a unidirectional transfer of information from the visual to the textual modality. Is there any possibility to conduct a bidirectional transfer?

**Limitations:**

See questions.

**Strengths And Weaknesses:**

The theoretical framework is clearly presented, and the experimental design looks sound. However, the proposed method is relatively simple and not verified in larger models such as Qwen3-VL-32B. And from the table 4, the performance on the TextVQA dataset is not satisfying, which needs further explainations. The paper is well structured but the paper is not easy to follow due to its high conceptual barrier and equations. I really suggest the authors to add more figures for better understanding. The question that the author addresses is quite interesting, but I would like to ask that how are the corrupted modalities created in your experiments and what is the practical usage of your method.

---

> ### Author Rebuttal · Authors · 2026-03-28
>
> We thank reviewer ms6p for recognizing that the work is presented clearly and for providing feedback. We address the concerns below:
>
> ### Q1: Explanations on TextVQA and Table 4.
> We ensured rigorous experimental controls across all redundancy levels, consistent SFT datasets, training parameters, and architectural configurations (Appendix C),  ruling out experimental error. We attribute the TextVQA results to two factors: first, TextVQA requires fine-grained visual grounding of text embedded in images, precisely the type of $u_V$ our method transfers away, creating a theoretically expected tradeoff consistent with our findings in Table 3; second, multimodal interactions affect different task categories differently (Sec 5.3), and the degree of this effect varies with model size and trainable parameters. We acknowledge that an exhaustive analysis of redundancy effects across all task categories is a promising direction for future work.
>
> ### Q2: Possibility of Bidirectional Transfer
> **Bidirectional transfer is feasible**. We conducted additional experiments on 1,000 samples from a vision-language sarcasm dataset (DocMSU), generating images conditioned on text using gpt-image-1.5. The table below shows that generated images nearly match real data in $R$ across all splits, effectively converting $U_T$ into $R$, mirroring our $U_V → R$ transfer. This serves as strong preliminary evidence that the MI Gate framework generalizes to bidirectional (Image $\leftrightarrow$ Text) interaction transfers.
>
> | **Split** | **Image Source** | **$R$**    | **$U_V$** | **$U_T$**  | **$S$** |
> | --------- | ---------------- | ---------- | --------- | ---------- | ------- |
> | Train     | Blank Images     | 0.0000     | 0.0000    | **0.6747** | -0.0001 |
> |           | gpt-image-1.5    | **0.6732** | 0.0034    | 0.0000     | 0.0152  |
> |           | Real-Data        | **0.6736** | 0.0147    | 0.0000     | 0.0039  |
> | Val       | Blank Images     | 0.0000     | 0.0000    | **0.5757** | 0.0032  |
> |           | gpt-image-1.5    | **0.4997** | 0.0089    | 0.0890     | 0.0278  |
> |           | Real-Data        | **0.5887** | 0.0766    | 0.0000     | 0.0000  |
> | Test      | Blank Images     | 0.0000     | 0.0000    | **0.5259** | -0.0152 |
> |           | gpt-image-1.5    | **0.5453** | 0.0347    | 0.0000     | 0.0506  |
> |           | Real-Data        | **0.5295** | 0.1409    | 0.0000     | 0.0061  |
>
> ### Q3: Clarifications on modality corruptions
> Modality corruption details are provided in Appendix D.2. Briefly, images are corrupted by progressively adding Gaussian, impulse, and shot noise; text is corrupted by progressively inserting, deleting, and replacing characters across 5 severity levels. In practice, our method applies to any real-world scenario where VLMs encounter noisy inputs, such as low-resolution images, OCR errors in scanned documents, or ambiguous queries; our hate speech detection experiment (Sec 5.1) is one concrete example. The MI Gate can be incorporated into any SFT pipeline as an offline preprocessing step with minimal additional overhead.
>
> ### Q4: Practical usage of the proposed methods
> Our method has two complementary practical applications. First, in real-world deployment scenarios where VLMs encounter noisy inputs, such as low-resolution images, OCR errors, or ambiguous queries, our method improves robustness by increasing exploitable redundancies in the training data before fine-tuning. Second, prior works (SMURF, RoBULT) have utilised redundant interactions in objective functions to train robust multimodal models; our MI Gate directly complements these approaches by increasing exploitable redundancies in the dataset before applying these objectives, addressing datasets with insufficient redundancy that would otherwise limit their effectiveness. To our knowledge, this is the first work to quantitatively adjust the degree of multimodal interactions within a dataset, offering a principled preprocessing step that supports and extends existing interaction-aware training approaches.
>
> Regarding larger models: our primary aim is to isolate the effects of multimodal interactions, which requires models with fully disclosed pretraining data to prevent confounding variables. Further, this work has partially addressed the issue of scale through the experiments with different model sizes in the same VLM family. Furthermore, training larger models with the extensive ablations and experiments involved in this work requires a very significant amount of computational resources. That said, we thank the reviewer for this feedback and will actively work on the analysis with larger models and include it in subsequent revisions.
>
> ### Q5: Additional figures for clarity and better understanding
> We thank reviewer ms6p for this suggestion. We will incorporate more figures in subsequent revisions for clarity and introduce the concepts in the paper to make it more accessible in Section 3 (e.g., visualization of $R$, $U$, $S$ in real data).

---

> > ### Author Rebuttal · Reviewer_ms6p · 2026-04-03
> >
> > 1. The author does not add additional figures for clarity and better understanding.
> > 2. For Table 4, the author says that "TextVQA requires fine-grained visual grounding of text embedded in images, precisely the type of $u_V$ our method transfers away, creating a theoretically expected tradeoff consistent with our findings in Table 3", I think a good method should also solve this problem.

---

> > > ### Author Response · Authors · 2026-04-04
> > >
> > > ### Q1: Additional figures for clarity.
> > > To clarify, we intend to add the supplementary figures in subsequent revisions. Following ICML’s submission policies, we are unable to upload a new revision during this discussion phase to reflect these updates.
> > >
> > > Nevertheless, we have uploaded a supplementary figure and its captions to an anonymized folder for the reviewer’s reference:
> > > https://imgur.com/HtX1M7A
> > >
> > > ### Q2: Further explanations on TextVQA
> > >
> > > In Table 4, we show that the architecture and size of the VLM affects how much redundant interactions can be learned or captured as it directly influences the number of trainable parameters (LoRA). Different models with varying levels of learnable parameters might require more (or possibly even less) redundant samples to maintain performance for different VL tasks.
> > >
> > > We can observe this in Table 4 where the performance for TextVQA in the largest tested (8B LLaVA) model dropped for the 25% additional redundancy, **but regained its performance at 50% additional redundancy**.
> > >
> > > There are other methods available that could achieve satisfactory performance on TextVQA. We wish to clarify that this is not the main focus of this work. We do not claim to provide or propose a better method to achieve higher performance on the TextVQA. Instead, this work focuses on **proposing multimodal interaction transfers as a promising direction to curate data and produce more robust vision language models**.
> > >
> > > The rationale for including the results of the general benchmarks is for completeness and to direct potential future works for tuning multimodal interactions for specific VL tasks.
> > >
> > > We hope these clarifications would clear up your remaining concerns; if they do, we also hope that you would kindly consider increasing your scores to reflect the resolved queries.

---

### Official Review · Reviewer_X5Ni · 2026-03-18

**Soundness:** 3
**Presentation:** 3
**Significance:** 3
**Originality:** 3
**Overall Recommendation:** 4
**Confidence:** 3

**Summary:**

This paper analyzes different types of multimodal interactions for MLLMs in VQA and found that increase exploitable redundant interactions in instruction turning data can mitigate hallucination and improve the performance. Then it presents a workflow to curate such data from existing datasets. According to the experimental results, using the proposed dataset would improve the results on some VQA benchmarks, including MMMU, MMStar, MathVista, and TextVQA.

**Compliance With Llm Reviewing Policy:**

Affirmed.

**Final Justification:**

The response have addressed my concerns.

**Key Questions For Authors:**

Weaknesses 1, 2, and 3.

**Limitations:**

yes

**Strengths And Weaknesses:**

Strengths
1. Clear motivation with rich analysis (e.g., sec 3.2). It is easy to follow the authors' ideas.
2. Simple but effective method. The data curation pipeline is simple (illustrated clearly in Figure 1) but effective in some VQA benchmarks validation.
3. Well presentation. There are many easy-to-follow analyses in the paper.

Weaknesses
1. Too many rule-based designs in the data curation pipeline. I think it is not scalable.
2. The improvement is limited and affected by hyperparameters significantly (e.g., redundancy level).
3. No validation on recent models (e.g., qwen-3-vl and internvl 3.5). They are much more powerful than LLaVa-OneVision. The proposed method may not work.

It is not a paper for state-of-the-art scores, but I think the analysis would bring some new insights to the community.

---

> ### Author Rebuttal · Authors · 2026-03-28
>
> We thank Reviewer X5Ni for their valuable feedback and for recognizing our clear presentation and proposed methods. We address the concerns below:
>
> ### W1: Scalability of the data curation pipeline
> The design choices in the MI Gate are theoretically motivated rather than empirically hand-tuned: the dominance criterion (Alg. 2) and synergy exclusion follow directly from the PID framework (Sec 3 and 5.1), not arbitrary rules. In practice, the pipeline was already applied to 984K samples across 48 datasets offline, demonstrating scalability comparable to standard instruction dataset curation. The only additional cost over standard SFT is the captioning step, which is trivially parallelizable. We also intend to release our codebase and curated datasets to enable further community improvements.
>
>
> ### W2: Effects of hyperparameters
> Rather than a sensitive tuning parameter, $\tau$ provides a principled knob to systematically control the degree of interaction adjustment. Fig. 3 shows a consistent improving trend across $\tau$ values (0–90%), demonstrating the method is not brittle; the failure at $\tau=100\%$ is expected (Sec 5.1, 5.3). On limited improvements: our primary contribution is robustness, where we demonstrate substantial gains, 38.3% reduction in visual-induced hallucinations and 16.8% improvement in consistency, rather than benchmark scores. Additionally, $\tau$ is a useful indicator for how much we can adjust the multimodal interactions within the dataset. This provides valuable early insights before processing the dataset, reducing the risk of excessive sunk costs
>
>
> ### W3: Validation on newer models
> Our primary aim is to isolate the effects of adjusting multimodal interactions, requiring models that are open-source in both weights and pretraining/fine-tuning data. Models like Qwen3-VL and InternVL 3.5 have not fully disclosed their pretraining data, introducing confounding factors that would make it difficult to attribute observed changes to redundancy alone. That said, our findings are consistent across two model families and five model sizes (256M–8B), providing strong evidence of generalizability across architectures and scales. We acknowledge extending to more recent models is a worthwhile direction and leave this to future work.

---

> > ### Author Rebuttal · Reviewer_X5Ni · 2026-04-04
> >
> > Some concerns hvae been addressed. But it is still unclear whether it can work well on rencent models.

---

> > > ### Author Response · Authors · 2026-04-04
> > >
> > > We would like to clarify three points that would address reviewer X5Ni’s remaining concern:
> > >
> > > 1. The LLaVa OneVision-1.5 model that we experimented on for this work was released around 6 months ago. At the point of our experiments, this model was one of the recently released open-data and open-weights VLM that we could experiment on.
> > >
> > > 2. The models tested in this work align with the training and architecture paradigms of the suggested VLMs (e.g., InternVL); they share the same fundamental design principles (e.g., a vision encoder and a modality projector to a LLM) with similar training objectives (e.g., pretraining $\rightarrow$ instruction tuning). As such, we reason that our approach is generally extensible to these more recent models.
> > >
> > > 3. Training on recent VLMs without open-data and open-weights **inherently introduces confounding factors** which compromises on scientific rigour. For instance, if the pretraining data is not disclosed, it would be difficult to attribute our results solely to the change in multimodal interactions — which is also the main focus of this work.
> > >
> > > We hope these clarifications would clear up your remaining concern; if they do, we also hope that you would kindly consider increasing your scores to reflect the resolved queries.

---

### Decision · Program_Chairs · 2026-04-30

**Decision:**

Accept (regular)

**Comment:**

This paper proposes a Self-Captioning Multimodal Interaction Tuning (MIT) workflow to address hallucinations and robustness issues in Vision Language Models (VLMs) caused by ambiguous or corrupted inputs. The Multimodal Interaction (MI) Gate decomposes data attributes into redundant (shared), unique (exclusive), and synergistic (emergent) information, and specifically identifies data points with high levels of unique visual information. Based on that, the framework transfers unique visual cues into redundant interactions through image captioning. Experimental results demonstrate that amplifying these redundancies significantly enhances model resilience against external noise, reducing visual-induced errors and improving response consistency.

Reviewers praised the analysis of systematically categorized data attributes effect on model learning, the theoretically-grounded pointwise information decomposition framework, and extensive experiments across tasks and model scales.

Reviewers expressed a primary concern that the method was not attempted on more recent models or larger models, questioning whether the method would still work on new models. The authors pointed out in the rebuttal that their model was recent at the reasonable time of conference manuscript preparation, and that repeating experiments on newer models where training data is not known may introduce bias.

On the premise of the generally positive reviews and remaining questions of the reviewers towards relevance of the model in light of recent public model releases, I recommend weak acceptance of the paper.